# Inner core backtracking by seismic waveform change reversals

Wei Wang[1,2,3], John E. Vidale[3✉], Guanning Pang[4], Keith D. Koper[5] & Ruoyan Wang[3]

The solid inner core, suspended within the liquid outer core and anchored by gravity, has been inferred to rotate relative to the surface of Earth or change over years to decades based on changes in seismograms from repeating earthquakes and explosions[1,2]. It has a rich inner structure[3–6] and influences the pattern of outer core convection and therefore Earth's magnetic field. Here we compile 143 distinct pairs of repeating earthquakes, many within 16 multiplets, built from 121 earthquakes between 1991 and 2023 in the South Sandwich Islands. We analyse their inner-core-penetrating PKIKP waves recorded on the medium-aperture arrays in northern North America. We document that many multiplets exhibit waveforms that change and then revert at later times to match earlier events. The matching waveforms reveal times at which the inner core re-occupies the same position, relative to the mantle, as it did at some time in the past. The pattern of matches, together with previous studies, demonstrates that the inner core gradually super-rotated from 2003 to 2008, and then from 2008 to 2023 sub-rotated two to three times more slowly back through the same path. These matches enable precise and unambiguous tracking of inner core progression and regression. The resolved different rates of forward and backward motion suggest that new models will be necessary for the dynamics between the inner core, outer core and mantle.

The inner core (IC) has been known to change over decades since the discovery of changing seismograms of repeating earthquakes[1,7]. The dominant interpretation of steady super-rotation over decades has been derived from temporal changes of up to tenths of a second in the difference in arrival times between PKIKP and later core phases in repeated earthquakes. The inferred rate of super-rotation has settled to about 0.05–0.15° per year, and motion in the past decade may have slowed[8–10]. Similar rates have been inferred from normal modes[11], PKIKP coda wave changes[12], IC-backscattered waves[2,13,14] and antipodal core waveform changes[15]. Fluctuating and much faster motion has also been suggested[16]. Most recently, observation with medium-aperture, high-frequency arrays and individual stations has suggested that PKIKP coda waves from 1991 to 2017 changed over time primarily during the interval 2001 to 2003, which is interpreted as because of 0.5° IC rotation during that period and much less rotation at other times[17].

Other studies suggest oscillating motion. The distinct six-year oscillation (SYO) in the length of day (LOD) could be explained by gravitational coupling of mantle density anomalies and core–mantle boundary topography with inner-core boundary (ICB) topography[18,19], although alternate explanations have been proposed[20–22]. A reversal of motion inferred from backscattered seismic waves was consistent with the amplitude and phase predicted from the SYO pattern of LOD oscillation[19,23].

Apparent inconsistencies with the pattern expected from rotation in changes in PKIKP coda have been argued to preclude interpretation of solid-body IC rotation, and instead indicate structural changes in the IC or at the ICB, or conceivably in the outer core (OC)[24–26].

To resolve the inconsistency of recent models, here we gather and analyse additional data sensitive to IC changes. We focus on two short-period, medium-aperture seismic arrays in northern North America, the Eielson (ILAR) and Yellowknife (YKA) arrays, which record IC-sensitive PKIKP waves from earthquakes in the South Sandwich Islands (SSI). We compile repeating earthquakes from the literature for 1991–2020, and crucially add 12 new repeating earthquakes for 2021–2023. We carefully examine the seismograms for changes in PKIKP and its coda. The dependence of waveform changes on earthquake pair dates is used to construct a new model for IC rotation.

We collected a dense sampling of repeating earthquakes (Fig. 1b). We focus on the region in which IC change was first noted[5], and which has clear waveform changes and changes in the time difference between core phases (ddt) over more than 50 years (ref. 27)—the path from the SSI to northern North America (Fig. 1a). This path is close to north–south, a bearing shown to be most likely to reveal waveform changes from IC rotation[17]. Beamforming greatly improves the signal-to-noise ratio, so we select the high-quality, 20-element ILAR and YKA arrays, which have been recorded for more than 20 years. They were designed with apertures and siting appropriate for capturing clear teleseismic P waves at periods near 1 s.

We compile 121 events from 1991 to 2023 (Supplementary Table 1) in 42 locations, including 16 multiplets (Supplementary Table 2) of three

[1]Key Laboratory of Earth and Planetary Physics, Institute of Geology and Geophysics, Chinese Academy of Sciences, Beijing, China. [2]College of Earth and Planetary Sciences, University of Chinese Academy of Sciences, Beijing, China. [3]Department of Earth Sciences, University of Southern California, Los Angeles, CA, USA. [4]Department of Earth and Atmospheric Sciences, Cornell University, Ithaca, NY, USA. [5]Department of Geology and Geophysics, University of Utah, Salt Lake City, UT, USA. ✉e-mail: jvidale@usc.edu

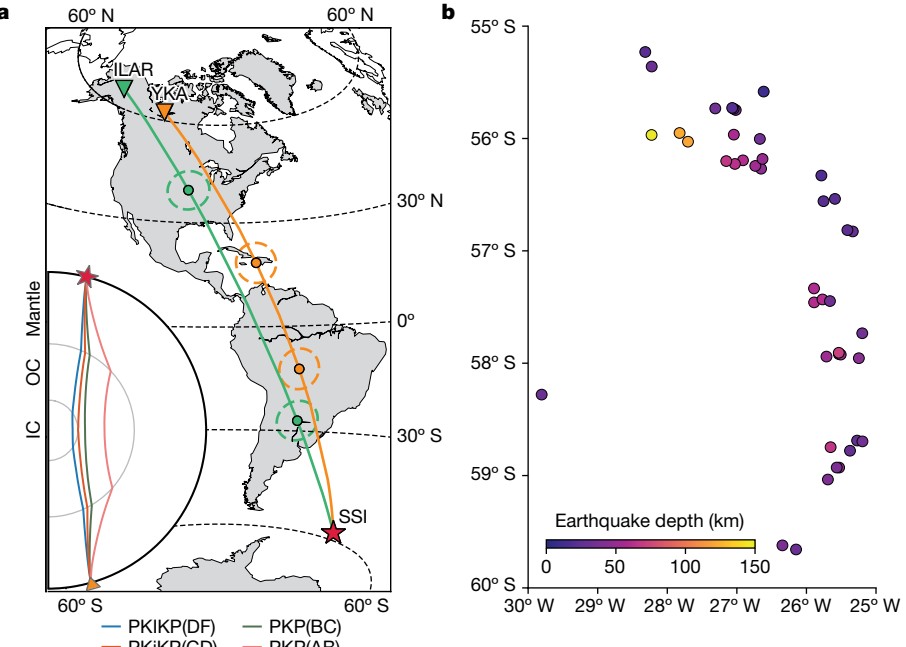

**Fig. 1 | Seismic ray paths and event locations. a**, Ray paths of PKIKP and PKP from the SSI source region to the two arrays (ILAR and YKA). The sampled IC region with a representative 1.5 Hz Fresnel zone[30] is marked with dashed circles centred at the PKIKP pierce points at the ICB. Inset, the ray paths of PKP (PKP(AB) and PKP(BC)), PKiKP(CD) and PKIKP(DF). **b**, Map of the SSI region with the source locations coloured by focal depth.

to seven events, which span 5° in latitude. The latest 12 events were found with a template search (Methods). These earthquakes form 143 pairs of repeating events (Supplementary Table 3). Between the two arrays, we made 200 waveform pair comparisons. The comparisons were done with stacks across each array (Methods).

Many PKIKP waves showed changes over the years, whereas we noticed no evidence that non-PKIKP phases changed in either arrival time or waveform, including IC-reflected phases. Many examples of these event pairs with changing waveforms have been presented in ref. 17.

We scored PKIKP by visual inspection for all event pairs from both arrays, classifying the waveform match as similar, somewhat similar or different, resulting in 57, 72 and 71 pairs, respectively. There were also 48 pairs too noisy to evaluate and 38 for which data from ILAR,

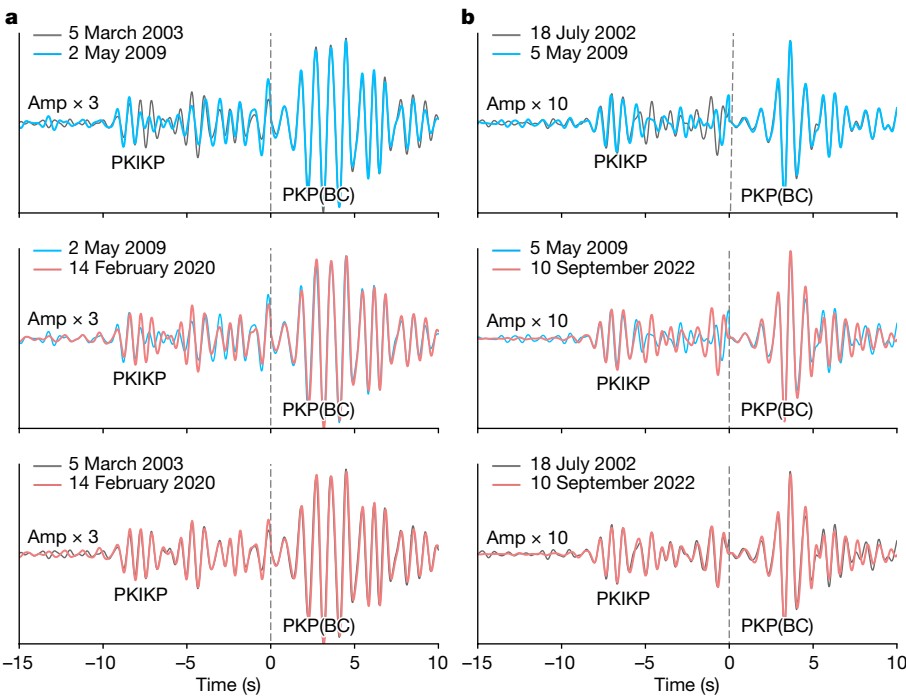

**Fig. 2 | Waveform comparison of multiplets. a**, The triplet that forms multiplet O, which repeats in 2003, 2009 and 2020. **b**, The triplet that forms multiplet J, which repeats in 2002, 2009 and 2022.

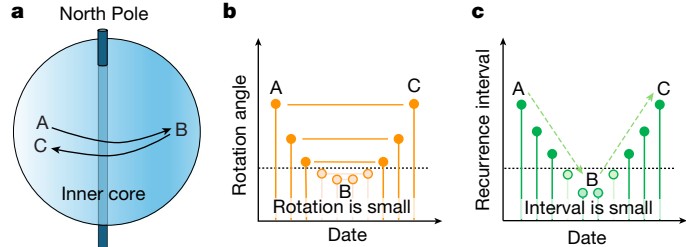

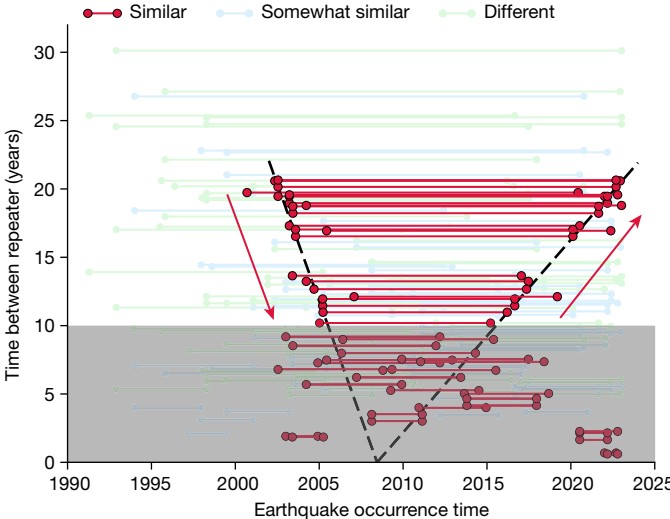

**Fig. 3 | Schematic of rotation observations. a**, Diagram showing the relation between rotation angle and recurrence interval at times A, B and C during reversal of differential rotation. **b,c**, Rotation angle (**b**) and recurrence interval (**c**) at times A, B and C during reversal of differential rotation.

**Fig. 4 | Dates of similar (red), somewhat similar (blue) and different (green) pairs of repeating events.** The dots are the years of events, the lines connect repeating pairs. The black line on the left shows that the trend of first events in a pair that has progressively shorter intervals occurs progressively later from 2000 to 2005. The black line on the right shows that the second event in a pair that has progressively longer intervals occurs progressively later from 2010 to 2023. Pairs less than 10 years apart are shaded, as just a few years of separation throughout the entire period apparently does not involve enough IC motion to always change the waveform. Lines from the ILAR array pair measurements are raised 0.4 years to visually separate them from the lines from the YKA array for the same event pairs.

which has the shorter archive, were not available. Almost all of our scores match the interpretation in ref. 17 for the presence or absence of waveform change. More objective scoring is possible[17], but some level of subjectivity would remain. Noise level, the time interval that the DF phase is above the noise, amplitude relative to nearby reference phases, character of other nearby repeating pairs and repeater similarity on global stations for non-IC phases all were evaluated, as well as potential differences between repeats in location and source time function. A further complication is that ILAR at 150° and YKA at 135° present PKP waves with distinct patterns of timing and amplitude of PKIKP and PKiKP, and interference with other core phases. The pattern and model described below become clear, in our opinion, and the model predictions should be testable within the next 5–10 years.

Figure 2 shows two examples of a triplet of event pairs constructed from two three-event multiplets. The middle-event waveform differs from those of the first and last events, which are essentially identical in each case. That is, remarkably, the PKIKP changes then reverts to the original across the three events. One or two such instances could simply indicate that the middle event is anomalous in a variety of possible ways, so we investigate more thoroughly.

The very similar initial first few seconds of most of the repeating-event waveforms is the expected result of scattering in a heterogeneous medium that has shifted. Waveform changes become greater with increasing lag time behind the direct arrival, as was demonstrated by synthetic seismograms in ref. 17.

The results for both arrays for all events, and just the 96 most similar events, are shown in Extended Data Figs. 1–4. Some broad patterns are evident. Pairs in the south show less difference at YKA. Most pairs that start in the early years change waveform. Note that there are fewer pairs for ILAR because of its later starting date for data availability from Incorporated Research Institutions for Seismology (IRIS). These observations are hard to translate into IC motion as plotted.

Notably, some widely separated pairs of events happen with unchanged waveforms, as noted in ref. 17. Even more surprising is that five or so multiplets, spread across the SSI region, change waveform and then change back across a span of a decade or more, as shown in Fig. 2.

We interpret below that these observations indicate a reversing IC that shifts first in one direction and then back to reoccupy the same position. Further examples of waveforms changing and reverting are shown in Extended Data Fig. 5. In this model, any event pair with matching waveforms at long intervals may well have produced different waveforms if a repeater had ruptured at times in between. Other pairs are similar but change in different pairings from the same multiplet with later or earlier times, as shown in Extended Data Fig. 6, and still others are simply pairs many years apart showing little change in SSI regions in which differently timed pairs generally do show a change. Southern SSI shows strong direct arrivals with weak scattered coda, with all changes more subtle, so we interpret waveform changes that are more subtle, notably in multiplets A and C. Scored changes for multiplet A are shown in Extended Data Fig. 7.

To investigate this model, we consider the dates of pairs with similar and different waveforms against their time separation. The matching pairs of times reveal when a rotation angle is repeated. In the context of previous models, which mostly find super-rotation in our early time span, probably the first repetition in matching repeaters is when the IC is super-rotating, and the later repetition is passing back through that same position while sub-rotating. The model and measurement are shown in Fig. 3.

The degree of similarity of the waveforms traversing the IC for all 143 most similar event pairs is shown in Fig. 4. The similar pairs tend to have their midpoint around 2010, with longer intervals of 15–20 years between events that extend farther from 2010, earlier for the first event and later for the second event. This is the pattern expected for an IC that has reversed direction near the date of the midpoint.

The pattern for all pairs, including the short recurrence times, showing which pairs do and do not fit this pattern, is shown differently in Extended Data Fig. 8. It is even more apparent there that for the matching pairs for longer intervals, the prediction in Fig. 3 matches closely the observations.

The shallower slope after 2010 in Fig. 4 indicates slower motion than before 2005, and projects to a reversal occurring in 2008 (Methods). We cannot resolve absolute rotation rate from this plot alone, the plot only measures the polarity and rate ratio between forward and backward rotations. Only asymmetry in rate across the time of reversal can generate the observed change in slopes.

The steeper slope before 2005 compared with that after 2015 shows that the IC motion is 2.5 times slower in the later period, as well as reversed (Methods). The IC motion has thus been more complicated than a symmetric function such as a sinusoid. We cannot trace motion back before about 2002—we see no waveform matches with events then, probably because the IC has not yet sub-rotated back to those positions.

The period between 2005 and 2015 is more difficult to resolve. We interpret that the rotation in this period slows as the IC position reaches an extremum before reversing. The time near the change in direction

produces a less definitive pattern of matches and mismatches, as slowing apparently lengthens the time interval over which the IC position remains similar. Waveforms across short intervals sometimes match even far from the turning point, also probably owing to only small changes in IC position.

There may be signs of more activity apart from just IC rotation; some pairs that the model predicts to match do not. More might be learnt from measuring time shifts in the changing waveforms and perhaps beamforming to locate and analyse individual scatterers that evolve between repetitions. Initial examination, not shown, suggests that ILAR ddt measurements do change and then revert in phase with waveforms. Here we simply present IC rotation with repeated waveforms and do not explore the pairs that should match but do not.

An IC that moves in one direction from 2002 to 2005, may not move much for a few years, then slowly backtracks from 2015 to 2023, resembles in broad form the recent years of motion in the model in ref. 27, which postulates a 70-year sinusoid slowing to reverse around 2010. Our measurements confirm the general trend, which had been controversial, and extend the observation period several more years, confirm a reversal and show asymmetry that had been not so clearly resolved. We verify for the first time that the path returns along a similar trajectory, without much wobble in the relative rotation pole.

Our model does not provide a strong test of the model in ref. 17, which suggests an earlier period of more rapid IC motion from 2001 to 2003, preceded and followed by much less motion. Here we see, however, that slow motion persists through most years since and measure its trend and relative speed. Repeating events for these and other source-station paths may well in future years start to match waveforms from still earlier times, elucidating the movement that generated the strong changes in waveforms that our study and ref. 17 observed for repeating pairs crossing the 2001–2003 window.

Our data do not resolve changes at the IC boundary or in the OC[26]; the PKP$_{BC/CD}$ arrivals do not change noticeably in waveform or timing. However, some earthquake pairs change when little is predicted from rotation and changes are seen when PKIKP and PKiKP overlap, allowing more IC variability than just rotation.

Our observations do not detect our previously favoured model of the mantle–IC gravitational coupling driving SYOs as the primary IC motion[23]. We note that the inferred change in polarity around 1971 (ref. 23), which is consistent with SYO predictions, is also consistent with the expected timing of an inferred previous reversal in the slow oscillation model[27] and a more variable rotation model[28]. We also note that improved estimates of the magnitude of IC motion necessary to cause the observed LOD oscillations include the likely entrainment of the OC in the tangent cylinder[29]. This additional inertial mass would reduce by a factor of two or three the angular amplitude of oscillation that would explain the SYOs in LOD, rendering it difficult to seismologically observe.

Our method and observations provide the most definitive evidence so far that the IC is moving relative to the rest of the Earth, and specifically that it is slowly and smoothly rotating on a reversing path. The observation that the westward sub-rotation is less than half as fast as the last part of the eastward super-rotation is well-resolved and begs models with that character. Identification of repeating pairs in which waveform changes and ddt from rotation cancel will allow greater resolution in the question of whether other processes near the IC boundary are also appearing. Examination with these methods of repeating IC waves on other paths, further in the past and into the future, promises rapid progress in monitoring motion in a difficult and enigmatic region.

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

# Article

## Methods

### Data selection and processing

We compile 109 previously identified events from 1991 to 2020, plus 12 new events from 2021 to 2023, in 42 locations (Supplementary Table 1). These events form 143 pairs of repeating events (Supplementary Table 2), including 16 multiplets (Supplementary Table 3) of three to seven events. With the 143 pairs and the two arrays, we made 200 waveform comparisons. The other 86 combinations lacked data (38) or were too noisy (48).

The 109 events are culled from the best compilations of repeating events in the literature[7,9,24,25,27,28].

These studies chose repeaters based on high waveform correlation for non-IC paths, variously above a cross-correlation of 0.90 or 0.95 over 15 s or more after the PKIKP onset.

We added pairs by connecting events across multiple lists and rejected several for having different source time functions, a few pairs because of high noise levels and one because the events had slightly different locations.

We searched data from 2021 to January 2023 for repeats of events already in the list compiled from the literature to add the crucial 12 later events, which led to 44 more repeating pair observations at the two arrays. We follow a criterion similar to previous compilations to find the additional 12 repeating earthquakes. The continuous data were searched only for events that matched a template in the original 109-event list. We use a 15-s time window to compute the cross-correlation (CC) of mantle phases and the non-IC phases $PKP_{BC}$ or $PKP_{AB}$ using the records on both dense arrays ILAR and YKA. The events with a median CC coefficient of more than 0.95 for both arrays are selected.

We further considered the degree of timing and waveform similarity in non-IC phases (mantle waves at other stations and $PKP_{BC}$ and $PKP_{CD}$ waves at the arrays) and divided the 143 event pairs (which form the 200 source–receiver combinations between the two arrays) into 96 more similar and 47 slightly less similar and/or slightly lower SNR event pairs (which yield 143 and 57 waveform comparisons, respectively; Supplementary Table 1).

The instrument responses are removed from the seismograms. Seismograms were transferred to velocity and filtered with a fourth-order Butterworth filter in the 1–2 Hz passband. We manually culled noisy stations and those with obvious clock errors, maintaining uniform sets of stations for each repeating pair. Then we simply normalized each trace and stacked each array with the slowness predicted for the IC waves in velocity model ak135 (ref. 31). We tested static corrections[13], but they made little difference, so we did not use them.

Slight location or source time function differences, perhaps masked by too much noise, could produce artefacts in patterns of waveform change that might mistakenly be attributed to IC changes. However, the results reported here do not vary much between including and omitting the 47 slightly less similar and lower SNR event pairs. We show all 200 repeating pairs in Extended Data Figs. 1 and 2, and only the more refined set of 143 repeaters in Extended Data Figs. 3 and 4.

To precisely align repetitions and identify waveform differences, it is important to accurately determine the time separation between repeating earthquake pairs. Our approach manually measures the difference in origin time based on the alignment of global records from other IRIS stations, assuming a common location for each event pair.

Estimating the slight differences in location between the earthquakes in each repeating pair, which we did not do, has been documented to be important in measuring the temporal change in the time separation of core phases (ddt)[32,33]. We avoid this complication by instead assessing waveform change, which is not sensitive to the very small time shifts that are the primary signal in ddt studies.

To ensure accuracy, we check the alignment of both the global records and the non-IC ILAR and YKA arrivals, namely, the PKP and core–mantle-boundary-scattered precursors. ILAR clock timing seemed flawless, but we noticed that YKA contained surprising clock errors. From 2013 to 2020, an evolving subset of YKA stations had previously unrecognized 0.125 s, 0.250 s or 0.375 s errors (N. Ackerley, personal communication), which were large enough to detect, estimate and correct (Supplementary Table 4). Also, a change in instrumentation in 2013 caused a 0.1 s jump[34]. There remain visible errors between events before 2013 on YKA of the order of 0.05 s, which we corrected by hand to align initial arrivals. As the network was timed with a single clock before 2013, these unexplained errors do not distort the stacked waveforms.

This rigorous process generally yields an absolute time accuracy of 0.03 s or better, facilitating comparisons in which there were emergent beginnings or changes in the initial waveform. As a side note, we were able to reproduce the ddt measurements for ILAR in ref. 27 well, but we do not show those ddt results here.

### Estimating the reversal time

In Fig. 4, the starting and ending times of the repeaters with similar waveforms (red dots) separated by more than 10 years show distinct linearity. We apply a linear fitting to each part separately, allowing intercept times $T_1$ and $T_2$ in equations (1) and (2) to differ.

$$t_{\text{beg\_pred}} = k_1 \times (t_{\text{beg\_obs}} - T_1) \tag{1}$$

$$t_{\text{end\_pred}} = k_2 \times (t_{\text{end\_obs}} - T_2) \tag{2}$$

$$\text{Misfit} = |t_{\text{beg\_pred}} - t_{\text{beg\_obs}}| + |t_{\text{end\_pred}} - t_{\text{end\_obs}}| \tag{3}$$

where $t_{\text{beg}}$ and $t_{\text{end}}$ are the starting and ending times of the repeaters in years, and $T_1$ and $T_2$ are the intersections at the $x$-axis. $T_{\text{pred}}$ is the predicted time intervals of the repeaters. $k_1$ and $k_2$, the two linear coefficients, together with $T_1$ and $T_2$, are the parameters for which we run the grid-search process to fit the observed recurrence intervals. The misfit function is shown in equation (3). We use the L1 norm instead of L2 norm to better ignore the outliers in the dataset. The uncertainties of the parameters are estimated using the bootstrapping method[35]. We randomly pick 15 (70%) from all the 22 data points, and run the grid-search process repeatedly for 1,000 times. The standard deviation is used to represent the uncertainty.

The best-fitting linear coefficients $k_1$ and $k_2$ are equal to $-3.54 \pm 0.27$ and $1.42 \pm 0.04$, respectively. The fitting intersection at the $x$-axis $T_1$ is equal to $2008.37 \pm 0.28$, and $T_2$ is equal to $2008.58 \pm 0.28$, so $T_1$ is almost equal to $T_2$. The lines are shown in Fig. 4. By contrast, we force the $T_1$ to be equal to $T_2$ and rerun the grid-search process. Our best fit shows a time of reversal of about $2008.5 \pm 0.18$, which is similar to that in ref. 27 using the PKIKP time shifts. And the linear coefficients $k_1$ and $k_2$ are equal to $-3.42 \pm 0.19$ and $1.42 \pm 0.03$. The lines are shown in Extended Data Fig. 9. It is noted that the slopes of the fitting lines are proportional to the rotation rate; consequently, we interpret that the rotation rate after $2008.45 \pm 0.19$ is about 2.5 times slower than that before 2008.5.

## Data availability

The seismic waveform data are available online from the Incorporated Research Institutions for Seismology Data Management Center (http://iris.edu) and the Canadian National Seismograph Network (http://earthquakescanada.nrcan.gc.ca/stndon/CNSN-RNSC/index-en.php). The events used in this study are listed in Supplementary Table 1.

## Code availability

All the codes will be available from the corresponding author upon request. All the figures were generated using Python packages, Matplotlib (https://matplotlib.org/), Basemap (https://matplotlib.org/basemap/stable/) and ObsPy (https://docs.obspy.org/).

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

**Acknowledgements** We thank M. Ishii, J. Aurnou, Y. He and J. Yao for their suggestions, and N. Ackerley for verifying and explaining the timing problem at YKA. We also thank A. Deuss, R. Deguen and other anonymous reviewers for their help. This study is supported by the National Science Foundation (grant EAR-2041892) and the Key Research Programs of the Institute of Geology and Geophysics, CAS (grant nos. IGGCAS-201904 and IGGCAS-202204).

**Author contributions** J.E.V. contributed to project design; W.W. and J.E.V. contributed to the methodology and data processing; R.W. conducted the numerical simulations to validate the IC rotation. All authors contributed to the interpretation of the observations and the preparation of the paper.

**Competing interests** The authors declare no competing interests.

**Additional information**
**Correspondence and requests for materials** should be addressed to John E. Vidale.

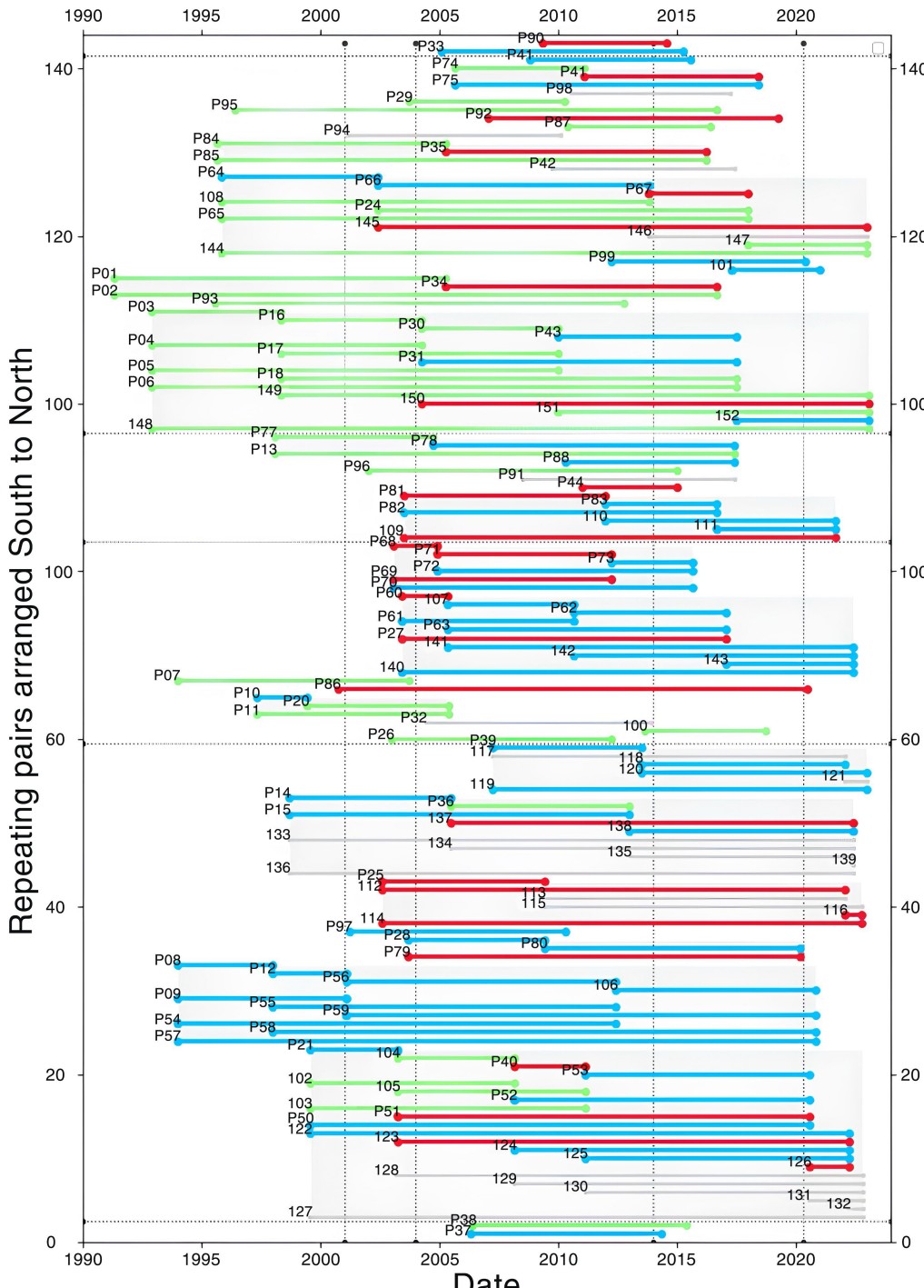

**Extended Data Fig. 1 | YKA waveform changes for all events.** The event pair indices key to Supplementary Data Table 2. Color indicates whether the PKIKP waveform changes between the two events in each earthquake pair. Red is similar, green is somewhat similar, blue is different, and gray is either too noisy or data is unavailable. The gray rectangles outline the multiplets with more than two events. The dotted lines mark the latitudes separating clusters of repeaters.

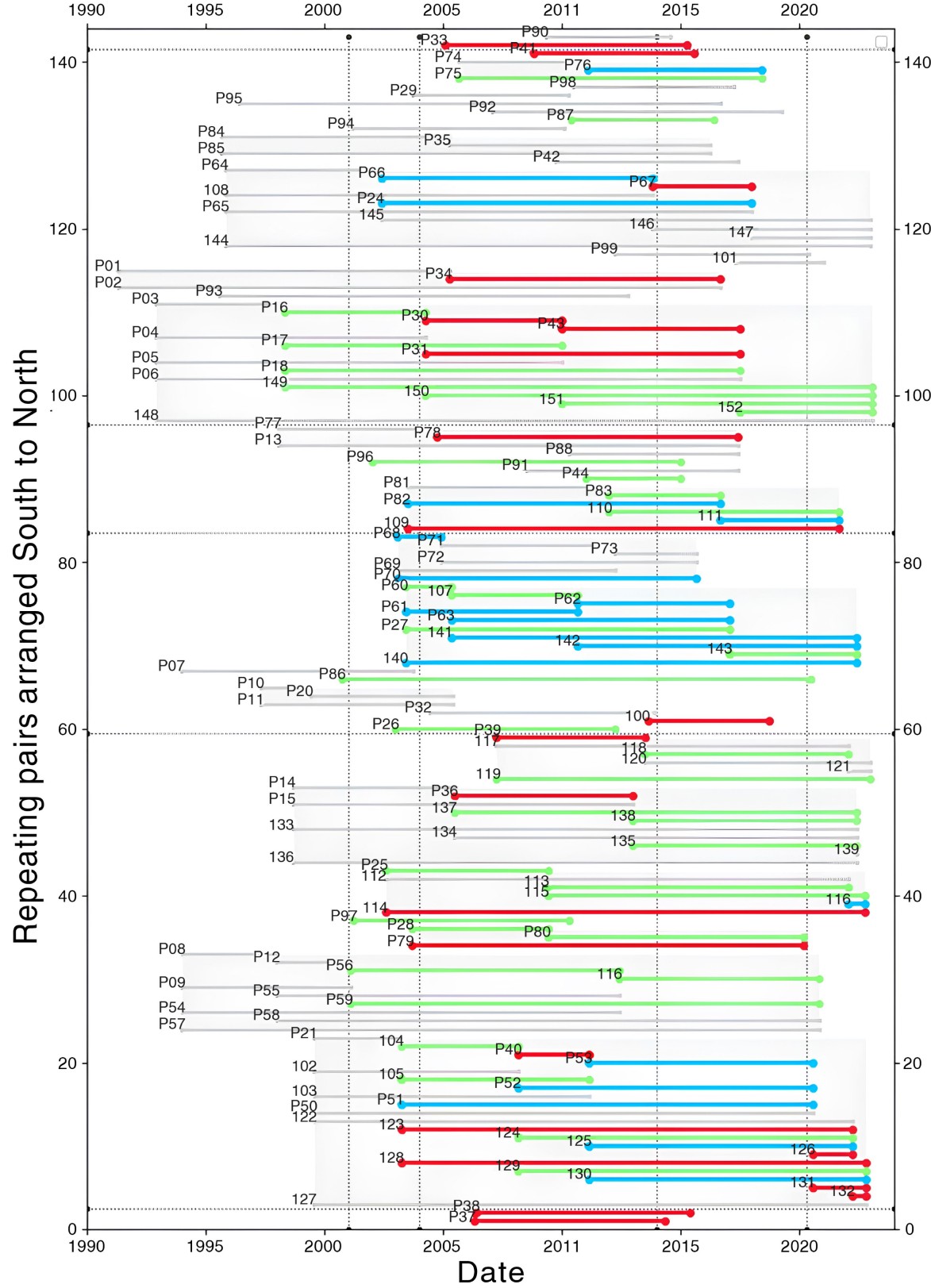

**Extended Data Fig. 2 | ILAR waveform changes for all events.** Similar to Extended Data Fig. 1.

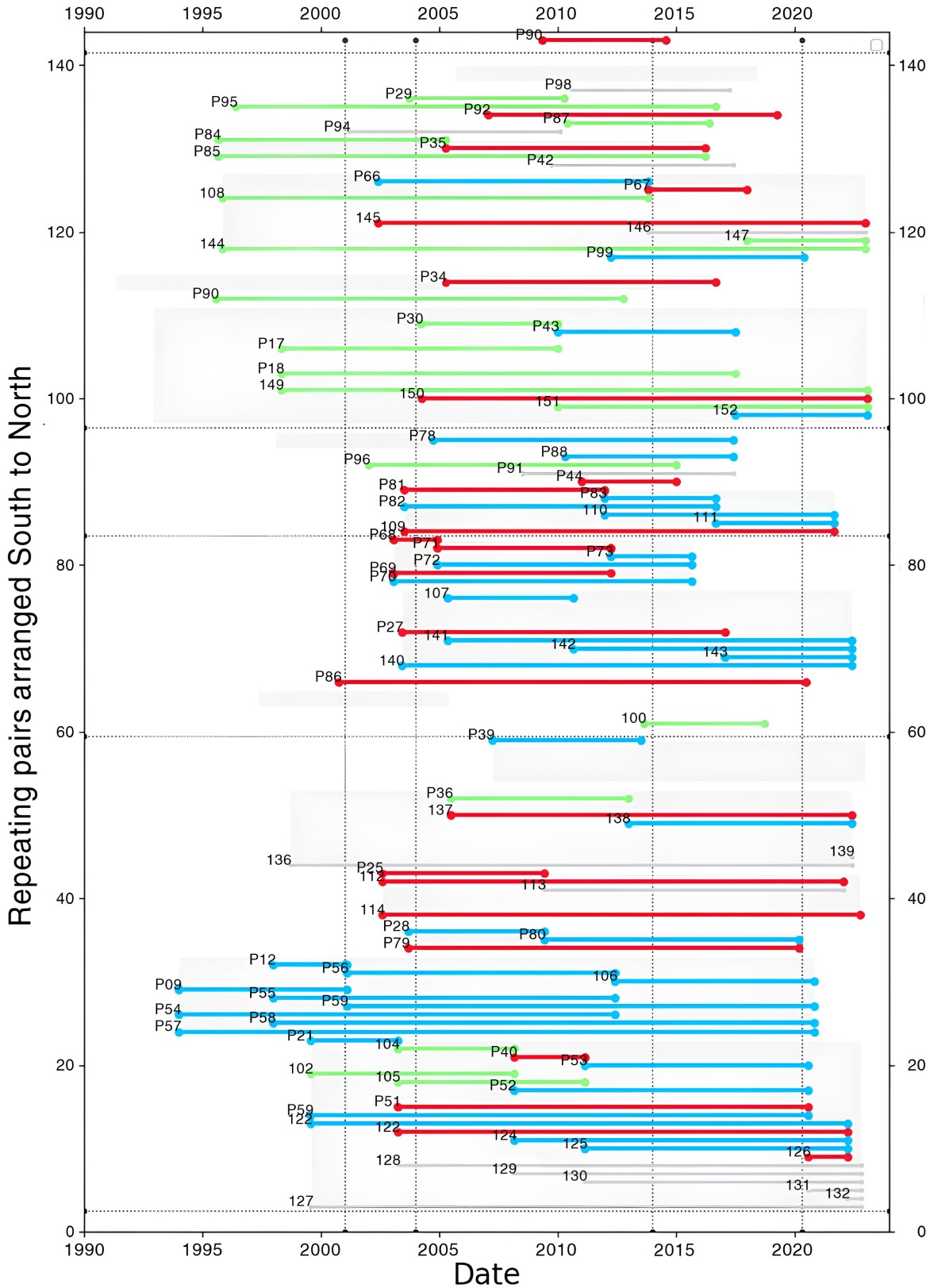

**Extended Data Fig. 3 | YKA waveform changes for the most similar events.** Similar to Extended Data Fig. 1.

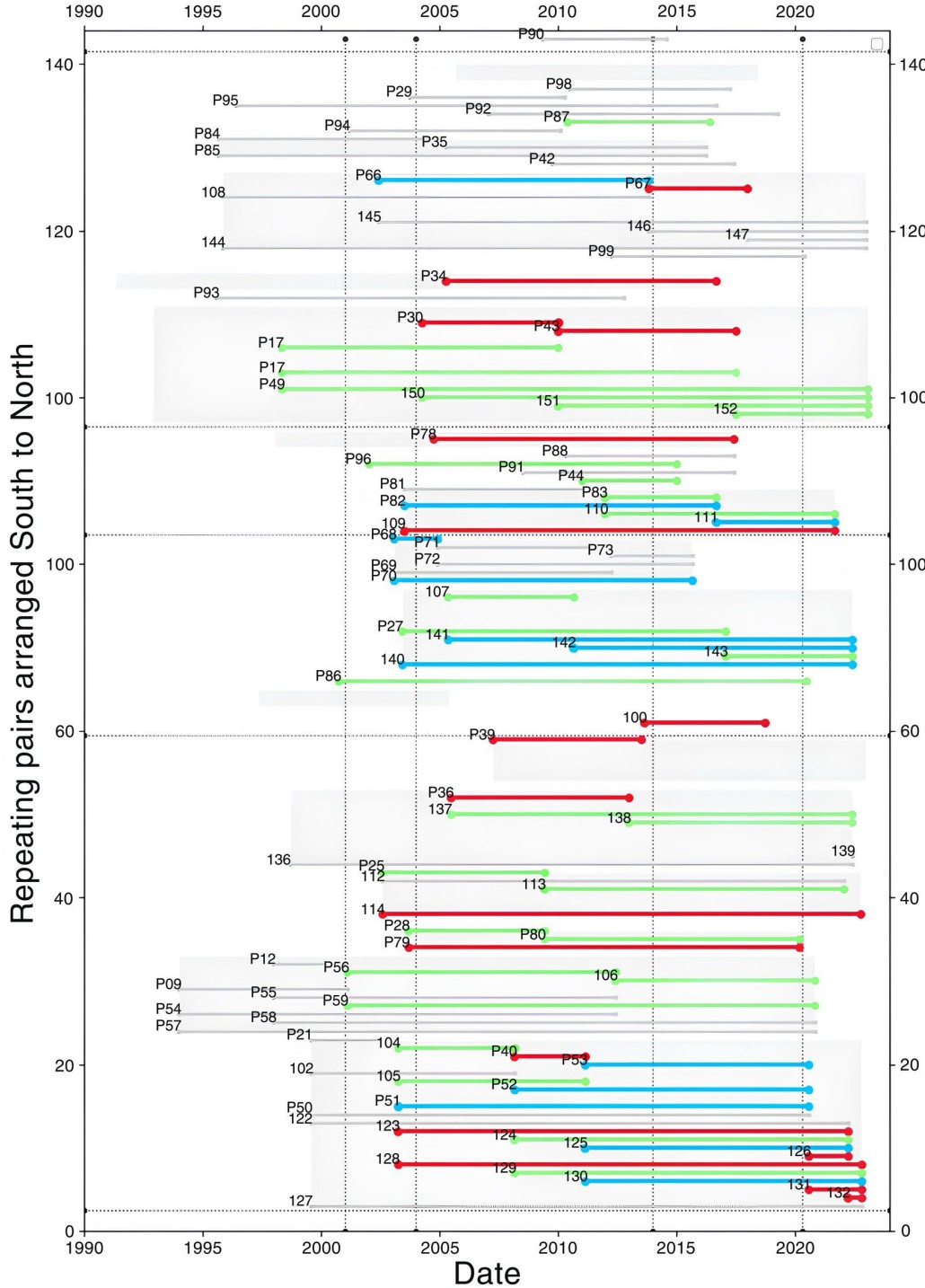

**Extended Data Fig. 4 | ILAR waveform changes for the most similar events.** Similar to Extended Data Fig. 1.

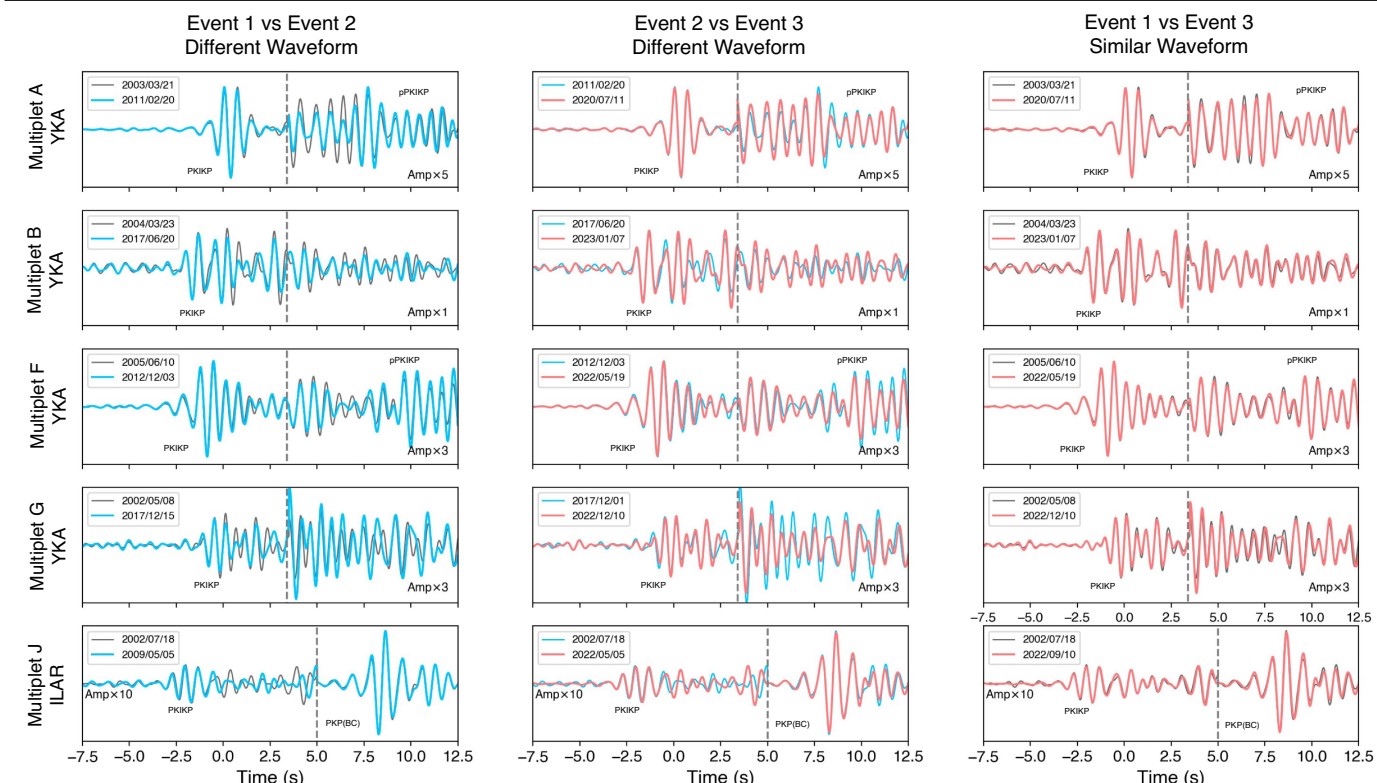

**Extended Data Fig. 5 | Waveform comparison of multiplets A, B, F, G, and J consisting of 3 events.** The left and middle columns show the variations in weak codas of PKIKP between the first and second events and second and third events, respectively. The right column shows the similar codas between the first and third events.

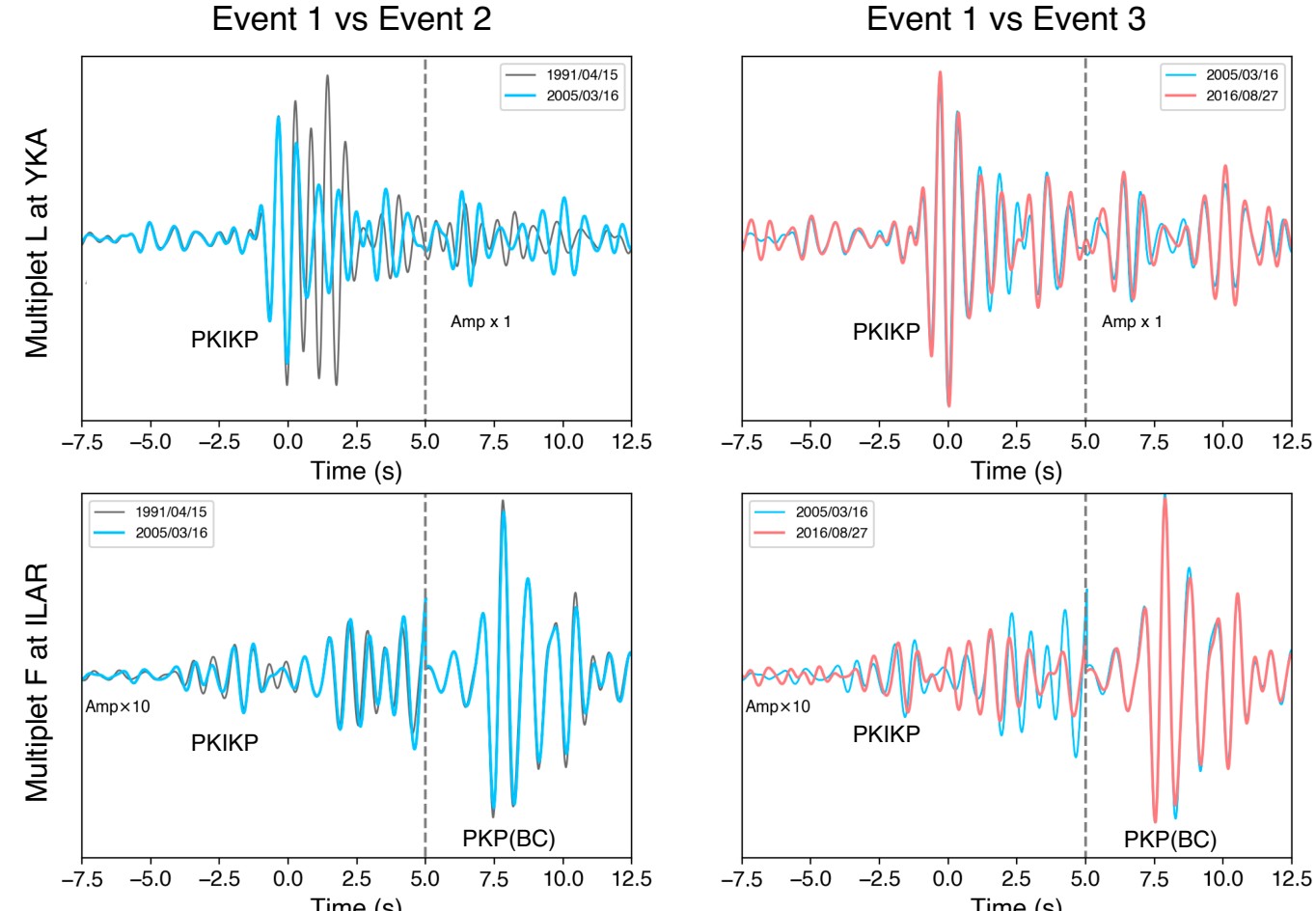

**Extended Data Fig. 6 | Waveform comparison of multiplets L and F.** (a) shows a big change, then little change across 1991, 2005, and 2016 in the multiplet L. Our inference is that the IC was in a similar position in 2005 and 2016. (b) shows a little change, then big change across 2005, 2012 and 2022. Our inference is that the IC was in a similar position in 2005 and 2012.

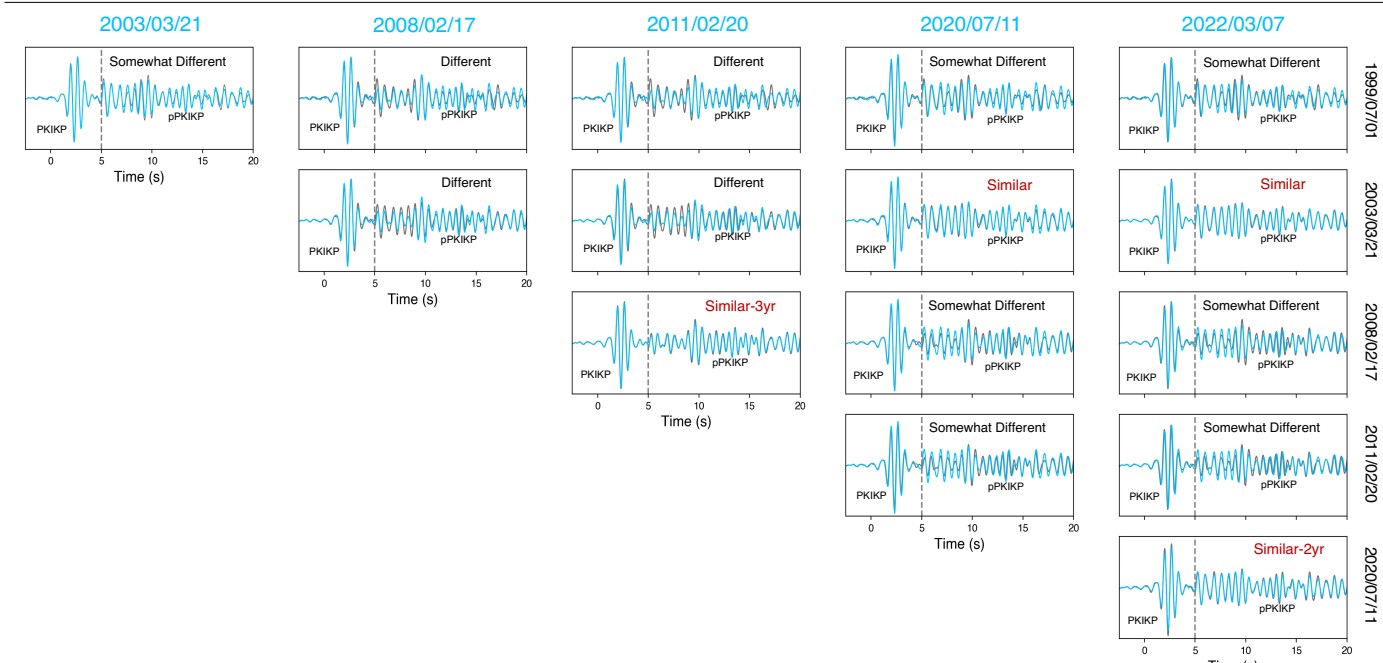

**Extended Data Fig. 7 | Waveform comparison of the events in multiplet A at YKA.** All the waveform comparisons for multiplet A are shown. Their location typically produces only subtle changes in waveform over time, which are particularly difficult to classify. The interpreted similarity or dissimilarity of each pair is labelled.

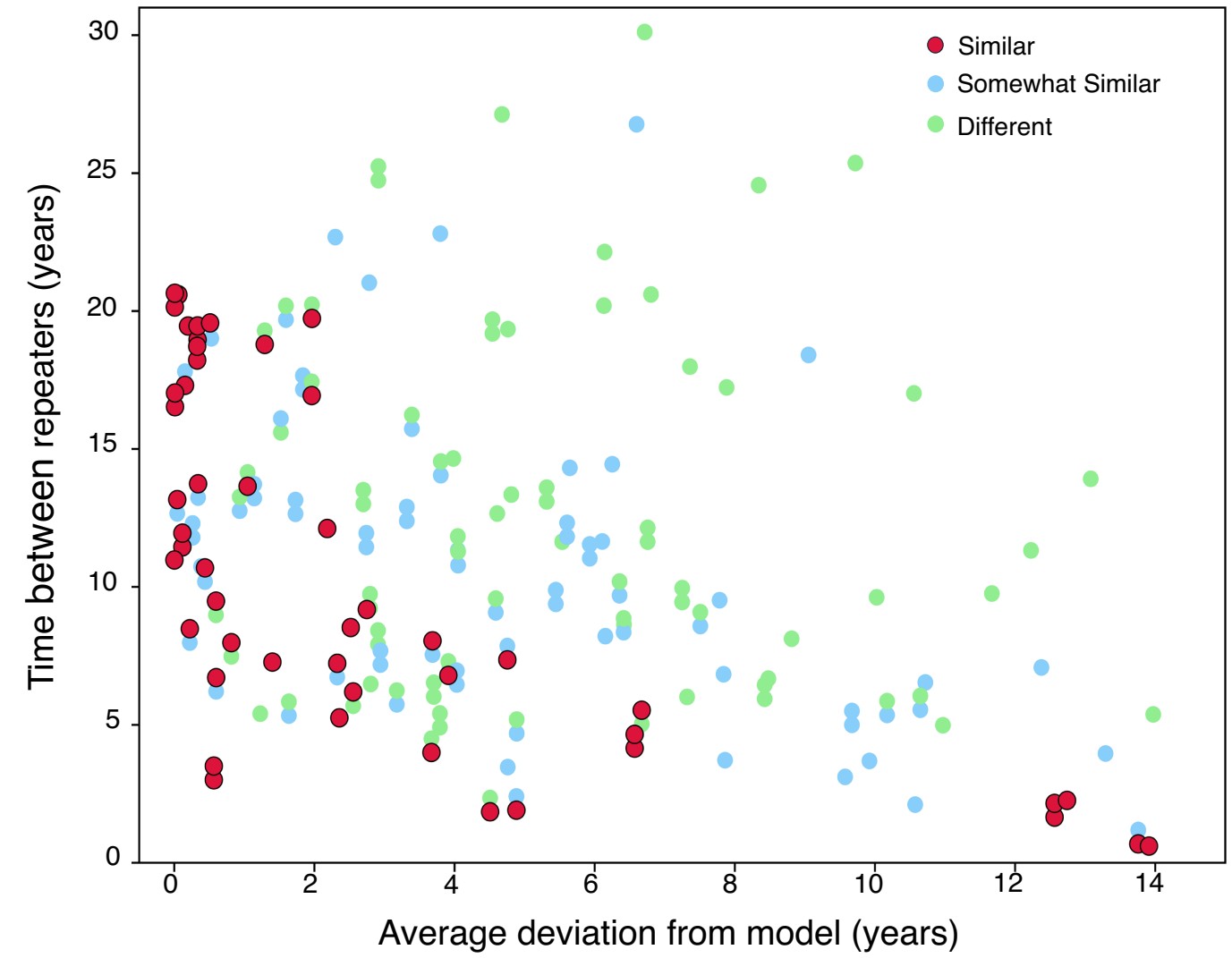

**Extended Data Fig. 8 | Degree of waveform change plotted on time interval between repeating events and average time distance from the predicted times of reversal.** Specifically, the distance is measured as the absolute value of the date of the first repeat minus the starting date predicted for the observed recurrence time, plus the absolute value of the date of the later repeat minus its predicted date. At intervals greater than 10 years between repeaters, most similar-waveform event pairs are much less than a year away from the pair of dashed-line predictions in Fig. 4.

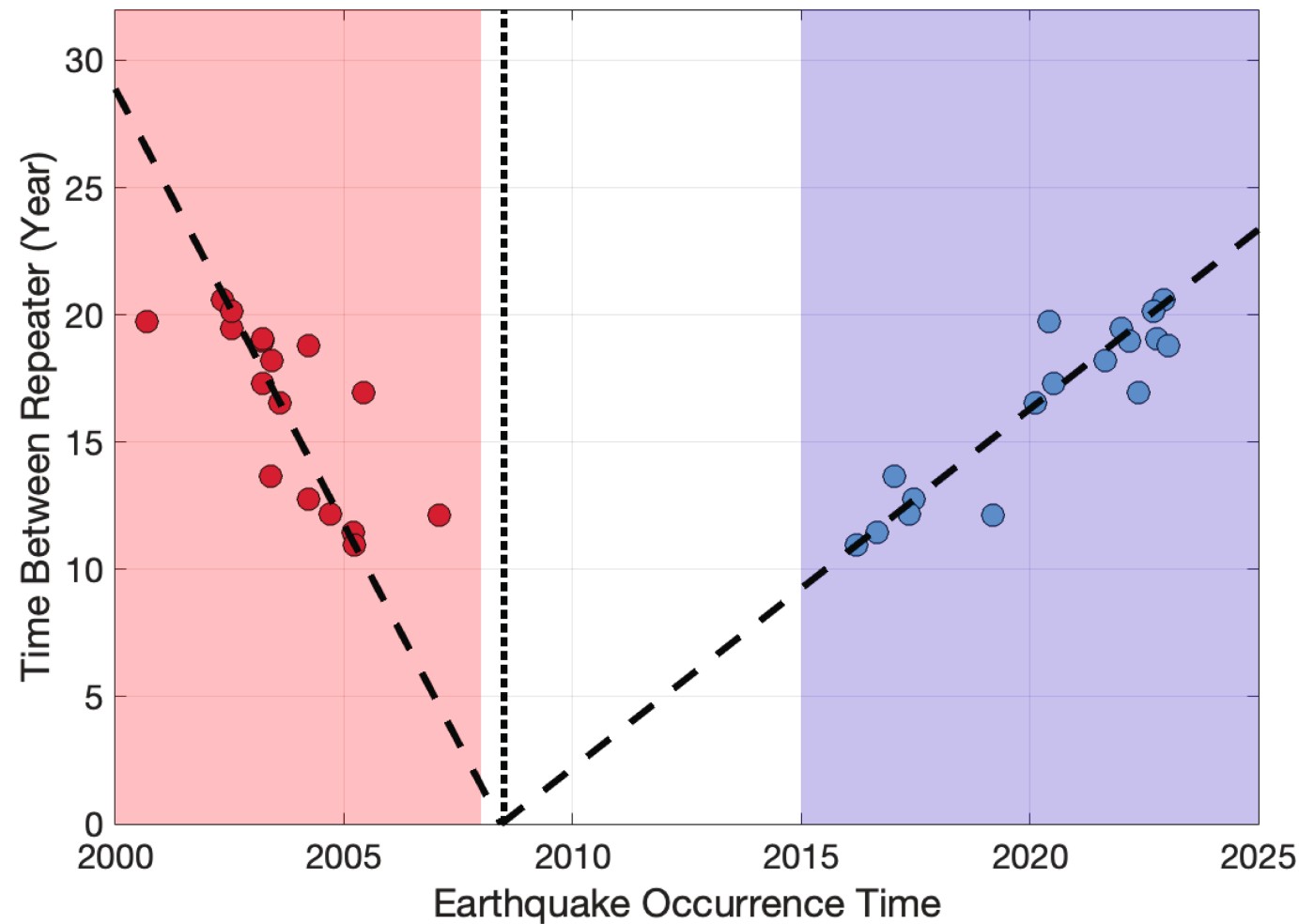

**Extended Data Fig. 9 | The red and blue dots mark the earlier and later events in similar-waveform repeating pairs.** The black dashed lines show the linear fitting with the lines constrained to meet at the x-axis, which are similar to Fig. 4. The light red area is the time when the inner core is seen to super-rotate, and the light blue area is sub-rotation. In between, matching pairs are less diagnostic, probably because inner core rotation slows as it is reversing, with minimal waveform changes.