## [Peer Review File · Nature]

Manuscript Title: Inner core backtracking by seismic waveform change reversals

Reviewer Comments & Author Rebuttals

Reviewer Reports on the Initial Version:

Referee #1:

I very much like the results on the inner core in which the authors show that the inner core moved forward and the backward, re-occupying the same position twice. I have seen the research being presented at conferences, and I am happy to see it is being submitted for publication in Nature.

It is the most carefully performed seismological study of a number of similar recently published studies on the same theme of the inner core oscillating back-and-forward or having a variable rotation rate instead of simply rotating in the same direction with a constant wave speed as was initially suggested.

I think the methodology is sound and the authors are very honest about what their data can and cannot resolve. I just have one question regarding the seismic data. When looking at their data in Figure 2, I did notice that in panel B, the first few swings of the PKIKP arrival (around -8 to -6 seconds) are in fact the same for all three events, and the main difference is in the train after the main PKIKP at -6 to 0 seconds. Why would the difference only be in the later part and not in the main PKIKP arrival?

I guess the main hesitation with the current paper is that the authors should much more clearly state how their results are new and different from previous work especially their own publication (Wang & Vitale, Sci. Adv. 2022) and Pang & Koper (2022) and Yang & Song (2023). In fact, I think these studies are very different and the results in the current manuscript are far superior compared to the other three papers. But that should be more obvious from reading the paper.

On a smaller note, I also would like to see some more discussion and comparison with normal mode estimates of inner core superrotation, which also had trouble finding a consistent rotation (work by Laske and others). Would it be possible to reconcile their proposed model with the normal mode estimates? Some other studies have suggested that the inner core may in fact only be rotating very slowly, at rates of 1 degree per million years or even less (work by Waszek and others). Would it also be possible to reconcile the proposed model with much slower rates over a long time?

Referee #2:

This manuscript presents PKIKP coda waveform changes (which is claimed by the authors as PKIKP waveform changes) recorded in two seismic arrays in Canada (YAK) and Alaska (ILAR). The authors claim to present the most definitive seismic evidence to date that the inner core has differentially rotated with

respect to the rest of the Earth and the inner core has backtracked with the time of the reversal being around 2008. They go on and infer the speeds of inner core differential rotation before and after 2008, and conclude that the inner core super-rotates from 2003-2008 and sub-rotates backward with a speed two or three times slower through the same path.

No information is given for any relocation procedures and any relocation results of the doublets. Those doublets are presented with the catalog information. The doublet source comes mostly from the databases of several research groups, with some newly identified by the authors. In the doublet sources, only those from Yao et al. (2009) had the relocation details and results. The source of the majority of the doublets in the list comes from Yang and Song's group. The doublets from Yang and Song's group were found to contain significant relocation errors (Zhang and Wen, 2023). The doublets need to be carefully relocated and need to be checked whether they are really close to each other. The temporal travel time changes need to consider the effects of the doublets' relative locations and origin times. Absent from the detailed analysis of doublet relative locations and their effects on the temporal change, the seismic results presented in the manuscript are questionable.

My following comments are based on the assumption that all the results stand after an analysis of doublet relocation. I do not see major changes in PKIKP waveform as the authors claim. Most of the changes occurred in the PKIKP waveform coda after the main phases (-1 s, -3 s and -7 s in Fig. 2a, and -2 s and -6 s in Fig. 2b). So, I would just use PKIKP coda changes in the subsequent comments. The distinction is needed as the PKIKP coda originate from a very different location of the inner core than the sampling region of the PKIKP waveforms.

Comment on the authors' definitive evidence of inner core backtracking -

The major evidence the authors claim that the inner core differentially rotates and backtracks to the original location is that the PKIKP coda change patterns between some triplets (three repeated earthquakes) and multiples (multiple repeated earthquakes). I use a triplet (events A,B,C, sequenced progressing in event occurring time) as an example for the discussion (Same logic applies to the multiples). The PKIKP coda exhibit changes from A to B, but no change from A to C. The authors interpret the change from A to B as the evidence that the inner core differentially rotates and, especially, no change from A to C as the inner core differentially rotates and backtracks to the initial position of A from B to C. The authors state that there is "the most definitive evidence to date that the IC is moving, and specifically that it is slowly and smoothly rotating on a reversible path."

PKIKP coda waveform changes in these arrays were well documented from the events occurring in the South Sandwich events (Figs. 2, S12-S16, S19-S22, Yao et al, 2019). Some sources of the PKIKP coda change were also well studied with a combined dataset from the two arrays used in this manuscript (YAK and ILAR) and three more seismic arrays (IMAR, BMAR, BCAR), with the waveform changes identified to be caused by a seismic scatterer near the inner core surface (Figs. 3, S17-1S18, Yao et al. 2019) that subsequently disappeared (Fig. 4, Session 3.1, Yao et al. 2019). Similar behaviors of the changing seismic scatterers were also well document in Yao et. al. (2015) that the temporal change of

the inner core surface is episodic, rapidly migrating, and alternately enlarged and shrunk.

The above-mentioned triplet waveform observations the authors presented can be simply explained by a mechanism that was well established in the observational evidence in the above literature: a disappearing scatterer. That is, an episodic change of the inner core surface appears around or before the time of event B and disappears after event B but before event C. The emergence of an episodic change of the inner core surface around or before event B would create waveform changes between AB, and the disappearance of the scatterer, or the recovery of the inner core surface, after event B and before event C would make waveforms dissimilar between BC and similar between events A and C. Or broadly, any localized waveform-changing scatterer that occurs around or before event B and subsequently disappears after event B but before C could explain the authors' observation. Inner core differential rotation is not needed. Inner core backtracking is just one of the possible ways to explain the data. It is not a piece of definitive evidence that the IC is moving or backtracking.

The authors' explanation of those triplet data as the inner core backtracking is contradicted by the seismic evidence presented in the manuscript as I will comment later. However, the rule of science would require the authors to address the following issues before continuing to present their favored hypothesis of inner core differential rotation.

For the objectivity of science, the past research results of Yao et al. (2019) and the above existing candidate interpretation need to be acknowledged. Alternatively, the authors should demonstrate the results in Yao et al. (2019) are not correct before claiming "the definitive evidence" for the inner core differential rotation. In the studies of the inner core differential rotation (same in this manuscript), Yao et al. (2019) was acknowledged providing an "alternative explanation". The temporal change of the inner core surface was shown to be a required interpretation for the observed temporal change of PKiKP phases in the earlier studies (Wen 2006, Yao et al. 2021). Yao et al. (2019) presented three lines of seismic evidence that are contradictory to the hypothesis of inner core differentiation. The subsequent published inner core rotation papers never addressed those three lines of contradictory evidence. Neither does this manuscript. Only one piece of contradictory evidence is needed to refute a hypothesis. The authors need to address the three lines of the contradictory evidence in Yao et al. (2019) before continuing to present their interpretation.

In addition, the authors need to demonstrate the theoretical plausibility of their favored interpretation. For authors' proposal to explain the data, following work needs to be done in order to make the proposal to be even theoretically plausible:

- 1) Authors need to identify the actual seismic structure that causes the temporal change of the coda waves from A to B.
- 2) Authors need to demonstrate that the identified static structure could cause the temporal changes of the coda waves based on synthetic modeling. As the Fresnel Zone is large for the seismic waves used in the study (see an example of the Fresnel Zone in Fig. 1 in Tian and Wen (2023)), a slight change of a static seismic structure would not be able to produce much travel time and waveform change. Previous

synthetic tests indicated that a static structure would need to be moved over a large distance to see the seismic effects, requiring an unreasonable super-super rotation of the inner core (see an example of synthetic tests in Fig. 8, Yao et al. 2019). The shortest time of waveform change observed in this manuscript occurs in a time scale of about 9 months (Fig. S2). The authors need to demonstrate that a moving of the static structure identified can produce the waveform changes observed.

While the inner core differential rotation studies have always illustrated proposals by beautiful cartoons, the above two points need to be synthetically tested. Before the above two points can be demonstrated, the inner core differential rotation cannot even be regarded physically plausible.

Comment on the contradiction of inner core backtracking to the actual data and the actual analysis in the manuscript -

The authors only analyze the data points of the doublets that exhibit no waveform change and have a time span of greater than 10 years, and discard those of the doublets that have a time span of less than 10 years and those that exhibit temporal changes. The authors state “as just a few years separation throughout the entire period apparently does not involve enough IC motion to always change the waveform”. There are two problems with this approach: 1) the statement is not justified and 2) even in the selected dataset of the analysis, there are many contradictory data points and those contradictory data points are discarded.

1) The statement “as just a few years separation throughout the entire period apparently does not involve enough IC motion to always change the waveform” is not justified based on the actual observations presented in the manuscript and the inferred time spans of doublet waveform change based on their conclusion.

From the actual observations:

Many doublets exhibit waveform changes when their time span is less 10 years. Just some examples from Fig. S1 only, these are the doublets that exhibit waveform changes but have a time span of less than 10 years: P41, P64, P99, 101, P43, 152, P88, PB3, P73, P10, 117, P14, P97, P28, P08, P12, P09, P2, P37. The shortest time span of the doublet that exhibits waveform change is event pair 116 in Fig. S2, which is 9 months and 8 days.

From the inferred time spans that must create waveform changes based on the author’s differential rotation interpretation:

I just note one in Fig. S1 for example:

P109 (Red, no change) (06/06/03 – 08/27/21) vs. P140 (Blue, change) (05/26/03 – 05/30/22)

P140 occurs 10 days earlier and about 9 months later than P109. As anything occurs in the time period

of P109 would be recovered based on the author's theory that the inner core backtracked to the original position, the waveform changes must occur in about 9 months in the time spans that the two doublets do not overlay.

The above observations and inference clearly show that the time scale of detectable temporal change in this dataset is as short as 9 months and that the author's discard of the data points of the doublets that have a time span of less 10 years is clearly unjustified.

2) Even in the selected data points of the doublets that have a time span of more than 10 years, which the authors presume representing the backtracking of the inner core to the initial position and use to infer the time of the inner core reversal and the super-rotation speeds before and after the reverse, many data points are directly contradicted by the observations in the other array. From Figs. S1 and S2, I list a few here:

P33: Red (no waveform change) in ILAR, Blue (waveform change) in YAK. In other words, ILAR indicates that the inner core backtracks to its original position and the data point was used to infer the backtracking (based on the authors' theory and analysis). But YAK indicates that the inner core does not backtrack to its original position as it exhibits waveform changes, and this contradictory data point is discarded.

P31: Red in ILAR, Blue in YAK.

P78: Red in ILAR, Blue in YAK.

P68: Red in YAK, Blue in ILAR.

P51: Red in YAK, Blue in ILAR.

All the above Blue contradictory data points are discarded in the analysis.

Putting back into analysis the data points of the doublets that have a time span of less than 10 years, we now do not see any consistent pattern of possible reverse of the inner core [e.g., Fig. S1]. Instead, we have various "modes" of inner core backtracking with the reversals occurring at different times and depending on which doublet data points you use. The data points for the reversal are contradicted by the waveform changes of other doublet data in the same time periods. We also see no evidence of reversal from the doublets occurring in the latitude range between P21 and P08 [Fig. S1] from YAK and that between 140 and P68 from ILAR [Fig. S2].

More contradictory data points emerge between the arrays in the doublets that have a time span of less than 10 years [Figs. S1, S2].

P41: Red in ILAR, Blue in YAK.

P43: Red in ILAR, Blue in YAK.

P39: Red in ILAR, Blue in YAK.

P116: Red in YAK, Blue in ILAR.

P31: Red in ILAR, Blue in YAK.

Clearly, the interpretation of the no temporal waveform change between the doublets or triplets as the result of inner core backtracking is not consistent with the seismic data, when we collectively and objectively analyze the seismic data. The seismic observations presented in the manuscript is a clear piece of seismic evidence that contradicts the hypothesis of inner core differential rotation.

In fact, the seismic results in the manuscript are also contradictory to the most recent high-profile conclusions of inner core differential rotation, including those of the authors of this manuscript.

Wang and Wei (2022) found that “the inner core subrotated at least 0.1° from 1969 to 1971, in contrast to superrotation of $\sim 0.29^\circ$ from 1971 to 1974” and proposed that the inner core oscillates in a 6-year cycle. Wang and Wei (2022)’s results were derived from the data points of the two groups of nuclear tests in a year span of 5 years, equivalent to one or two data points in Figs. S1 or S2. No 6-year cycle signal can be found in the 33-year time span of the data in Figs. S1 and S2. If we randomly select two sets of the doublets in Fig. S1 or Fig. S2 (like one nuclear test group from 1969-1971 and the other nuclear group from 1971-1974”) and presume the inner core differential rotation as the interpretation, we can derive many “modes” of inner core differential rotation that are contradictory to the bulk of the observations.

Pang and Keith (2022) concluded “We find that the inner core was nearly locked to the mantle before 2001 and after 2003 with relatively small motion about an equilibrium position. During 2001-2003, the inner core experienced a burst of differential rotation”. While their results were derived from a dataset that is also used in this manuscript, their conclusion is directly contradicted by the many BLUE doublet observations that have no time overlaps with 2001-2003 [Fig. S1] before 2017 (the year Pang and Keith (2022)’s doublet database ends).

Yang and Song (2023) concluded “that all of the paths that previously showed significant temporal changes have exhibited little change over the past decade. This globally consistent pattern suggests that differential inner-core rotation has recently paused.” That conclusion is directly contradicted by the many BLUE doublet observations that started after 2010 (when the last decade starts).

The reason that different groups or same groups at different publications obtained inconsistent results is that those inner differential rotation results were inferred from the seismic signals that are not related to inner core differential rotation and are intrinsically of another origin. Those seismic signals come from localized episodic temporal changes of the inner core surface that behave differently in differential geographic locations and in different time periods. They exhibit different temporal and spatial characteristics among differential seismic datasets. It is the misinterpretation of the seismic signals of the episodic localized changes of the inner core surface, coupled with the ignoring of the contradictory evidence, that resulted in many reported contradictory “modes” of inner core differential rotation.

It is true and unexplainable that the papers of the inner core differential studies could ignore the contradictory evidence in the literature (papers after Yao et al. 2019), discard contradictory observations from the analyses (Yang and Song, 2022 vs. Tian and Wen, 2003), and make statements likely

“unequivocally” without consideration of the alternatives and its contradiction to the other data (Yang and Song, 2022 vs. Tian and Wen, 2023). They could also discredit the other competing proposal with “erroneous claims” (Yang and Song, 2020 vs. Yao et al., 2021) or unreproducible results (Yang et al. 2021, vs. Zhang and Wen, 2023). Similar approaches are also adopted in this manuscript as I comment above. Another example is the statement in this manuscript that “Many PKIKP waves showed changes over the years, while no non-PKIKP phases resolvably changed in either arrival time or waveform, including IC reflected phases.”. While no results of doublet relocation were presented, the data experienced manual shifts, no identification of PKiKP phase was shown, and no mention of how the separation of PKIKP and PKiKP was made in the data, yet the authors make such a definitive statement that no change in arrival time in the IC reflected phases.

It is also true that high-profile publications of inner core differential rotation and “exotic behaviors” of the differential rotation would capture one wave of media/public attentions after another, as demonstrated in the past publications.

However, the ultimate outcome of ignoring the contradictory evidence, excluding the data that do not fit the hypothesis, and making a proposal without support of the physics of wave propagation will be the erosion of seismology as a credible branch of science and the destruction of seismologists as credible researchers.

In summary, while this manuscript presents interesting observations of waveform recovery across some doublets and a good dataset of temporal change of PKIKP coda waves, the interpretation of inner core differential rotation and backtracking is not supported by the bulk of the seismic observations presented in the manuscript. The subsequent results of the inferred inner core differential motions were obtained based on unjustifiable exclusions of the seismic data that do not support the hypothesis. The manuscript also failed to address the contradictory evidence to the hypothesis that was presented in the literature and in the manuscript, and the theoretical plausibility of the hypothesis. For these reasons, I do not recommend this manuscript for publication.

While it is not in my role as a referee to offer my view on the correct interpretation of the seismic data presented in this manuscript, I venture anywhere in case it is helpful. I think the authors have a unique dataset that sample one of the regions with intensive episodic changes of the inner core surface. Figs. S1-S2 clearly capture the spatial and temporal changes of a portion of the inner core surface that vary greatly in location and in time. A simple explanation exists to this seemingly contradictory and complex dataset. Each doublet captures a temporal change behavior at a particular spot of the inner core surface in the region during the occurring times of the doublet. A doublet with temporal change captures a temporal change of the inner core surface, a triplet or multiple with recovering waveforms reveals a temporal change of the inner core surface that has subsequently recovered, a doublet with contrasting temporal change patterns between the arrays (no change in one array and change in the other) represents a temporal change of the inner core surface that is more sensitive to one array and less sensitive to the other (due to location or the style of the change), and a doublet without change represents either a temporal change of the inner core surface that has subsequently recovered or no

change of inner core surface. The dataset is a clear piece of observational evidence that contradicts the hypothesis of inner core differential rotation and is a unique set of seismic constraints that could illuminate a detailed picture of episodic spatial and temporal changes of a particular region of the inner core surface. One way or the other, I am confident that this correct version of the interpretation will be published in a scientific journal.

Wen, L. (2006). Localized Temporal Change of the Earth's Inner Core Boundary, *Science*, 314, no. 5801, 967-970, doi: 10.1126/science.1131692.

Yang, Y., and X. Song (2020). Origin of temporal changes of inner-core seismic waves, *Earth Planet. Sci. Lett.*, 541, no. 116267, doi: 10.1016/j.epsl.2020.116267.

Yao, J., D. Tian, L. Sun, and L. Wen (2021). Comment on "Origin of temporal changes of inner-core seismic waves" by Yang and Song (2020), *Earth Planet. Sci. Lett.*, 553, no. 116640, doi: 10.1016/j.epsl.2020.116640.

Yang, Y., and X. Song (2022). Inner Core Rotation Captured by Earthquake Doublets and Twin Stations, *Geophys. Res. Lett.*, 49, no. 12, e2022GL098393, doi: 10.1029/2022GL098393.

Tian, D., and L. Wen (2023). Comment on "Inner Core Rotation Captured by Earthquake Doublets and Twin Stations" by Yang and Song, *Geophys. Res. Lett.*, 50, no. 15, e2023GL103173, doi: 10.1029/2023GL103173.

Yang, Y., X. Song, and A. T. Ringler (2021). An Evaluation of the Timing Accuracy of Global and Regional Seismic Stations and Networks, *Seismol. Res. Lett.*, 93, no. 1, 161-172, doi: 10.1785/0220210232.

Zhang, X. and L. Wen, (2023). Problematic Reported "Prevailing Clock Errors in Seismic Stations": Comment on "An Evaluation of the Timing Accuracy of Global and Regional Seismic Stations and Networks" by Yang et al. (2021), <https://agu.confex.com/agu/fm23/meetingapp.cgi/Paper/1349595>.

Yao, J., L. Sun, and L. Wen (2015). Two decades of temporal change of Earth's inner core boundary, *J. Geophys. Res.*, 120, no. 9, 6263-6283, doi: 10.1002/2015JB012339.

Yao, J., D. Tian, L. Sun, and L. Wen (2019). Temporal Change of Seismic Earth's Inner Core Phases: Inner Core Differential Rotation or Temporal Change of Inner Core Surface?, *J. Geophys. Res.*, 124, no. 7, 6720-6736, doi: 10.1029/2019JB017532.

Wang, W., and J. E. Vidale (2022). Seismological observation of Earth's oscillating inner core. *Sci. Adv.*, 8(23), eabm9916.

Pang, G., and K. D. Koper (2022). Excitation of Earth's inner core rotational oscillation during 2001–2003 captured by earthquake doublets. *Earth Planet. Sci. Lett.*, 584, 117504.

Yang, Y., and X. Song (2023). Multidecadal variation of the Earth's inner-core rotation. Nat. Geosci., 16, 182–187.

Referee #3:

The paper proposes a new way of using seismic multiplets to constrain the rotation of Earth's inner core (IC).

The analysis suggests that the rotation of the IC with respect to the mantle has changed direction around 2008. The authors base their conclusions on a detailed study of the waveform of seismic waves travelling through the IC (PKIKP), from a series of repeating South Sandwich Islands earthquakes (multiplets) from 1993 to 2023. The crucial observation is that in a number of multiplets, the waveform is observed to change with time before reverting to match the waveform of earlier events. The events with similar waveforms are interpreted as indicating that the IC has the same orientation with respect to the mantle, while different waveforms indicate different IC orientations. The analysis of all observed pairs of similar waveform events show a consistent pattern suggesting a change in IC rotation direction around 2008, as well as different rotation rates before and after 2008.

The results have important implications for the understanding of inner core dynamics and interaction with the core and mantle. The motion of the IC with respect to the mantle has been the subject of much debate. Robust observational constraints on IC rotation can provide key informations for understanding angular momentum exchange between mantle, outer core, and inner core, and provide constraints on the mantle/inner core gravitational coupling and IC effective viscosity.

This is to my knowledge the first time that the waveform of IC sensitive multiplets are analysed and used in this way. I find these observations quite exciting, and their analysis elegant.

As far as I can judge (I'm not a seismologist), the seismological observations seem robust, but I will leave this question to more knowledgeable reviewers. The model is simple (this is one of the strengths of the paper), and the assumptions behind it are clear for the most part.

The paper is clearly written and I have very much enjoyed reading it.

Apart from the suggestion detailed in my first point below, I have only very minor comments.

(1) While reading the paper, I have been wondering about how likely it is to sample the same IC orientation at two times picked at random, and about whether looking at the observations presented in the paper from a probabilistic point of view might give any useful information.

As an illustration, I did the following simple calculation, which perhaps will be of some use to the authors.

Let's denote by ϕ the rotation angle of the IC with respect to some reference, by $\Delta\phi$ the difference of

orientation of the IC between two events, and by $\Delta\phi_c$ a 'correlation angle' defined as being the change of ϕ below which no noticeable change in waveform can be found (i.e. the two waveforms are similar if $\Delta\phi < \Delta\phi_c$). If multiplied by the radius of IC, $\Delta\phi_c$ would give a length scale characteristic of IC internal structure (a 'correlation length').

It is found by the authors that roughly one fourth of the studied doublets happen to have similar waveforms (57 similar out of 200 doublets).

This could be interpreted as meaning that the probability, when taking a pair of observation times at random, of sampling the same orientation ϕ within an interval of width $\Delta\phi_c$ is about 25% ($\sim 57/200$). Or, equivalently, that there is a probability of about 25% to find $\Delta\phi < \Delta\phi_c$ for a random pair of events. Maybe some useful information can be extracted from this observation.

To be explicit, I have assumed that the angle ϕ evolves with time t as $\phi = \phi_0 (t/T)^2$ for t between $-T$ and T , where $2T$ is the length of the observation period, and ϕ_0 the amplitude of change of ϕ over this period. $t=0$ corresponds to the reversal time. This is a 2nd order Taylor expansion around the reversal time, assuming a symmetrical behaviour around it (it can easily be modified to account for the observed asymmetry around the reversal time). If we assume the rotation rate of the inner core to be ~ 0.1 deg/year, and take $T \sim 15$ years to be consistent with the data of the paper, one gets $\phi_0 \sim 1.5$ degree.

With these assumptions and a simple python script, I have drawn a random set of N pairs of times within the observation period, calculated the difference of orientation $\Delta\phi$ for each pair of times, and then calculated the cumulated density function (CDF) of $\Delta\phi$. The resulting CDF shows that the 25% of time pairs having the smallest orientation difference are such that $\Delta\phi/\phi_0$ is smaller than about 0.1. In other words, 25% of events pair would have similar waveforms if the correlation angle $\Delta\phi_c$ is about $0.1 \cdot \phi_0$. With $\phi_0 \sim 1.5$ degree, this gives $\Delta\phi_c \sim 0.15$ degree. Converting this into a length scale by multiplying by the radius of the inner core gives a correlation length of ~ 3 km.

This estimate clearly depends on the assumed form of the $\phi(t)$ function, on the parameters values I used, and, crucially, on whether the number of analysed event pairs is large enough to give a reliable estimate of the probability of finding similar waveforms. But at least the value obtained for the correlation length is not crazy, which to me gives additional support to the author's interpretation of their observations. And perhaps it is worth investigating further to check whether the value I have obtained for the correlation length is robust. If so that would give an additional measure of IC heterogeneity.

Anyway, this is really just a suggestion, which the authors are free to follow or not.

(2) Another point, which is somewhat related to point (1):

I would think that the interpretation of the data discussed in the paper depends to some extent on assumptions on the spatial structure of IC heterogeneities. For example, if the IC is overall homogeneous with only very localised heterogeneities, then finding twice the same waveforms should be the norm, and would not necessarily indicate that the IC is in the same position. I would therefore think that an

implicit assumption of the interpretation is that heterogeneities are evenly distributed, or that the IC properties vary gradually within the IC. I guess this interpretation is supported by the observations and others, but maybe this should be discussed?

Additional remarks :

- line 20-21: 'The pattern of matches shows that the IC super-rotated from 2003 to 2008, and then from 2008 to 2023 sub-rotated'

If I understand correctly, the observations presented in the paper do not allow to differentiate between super-rotation and sub-rotation (in fig 3, taking fig 3b upside down, i.e. having phi increasing and then decreasing, has no effect on figure 3c) but only show that the polarity of motion has changed. The interpretation in terms of super-rotation and the sub-rotation comes from the literature. The sentence of the abstract should make this clear, either by only mentioning the change of polarity, or by mentioning the use of previous results from the literature.

- line 47: the part of the sentence in between parenthesis ('and mostly rotation only during') is not clear to me.

- line 58: I believe 'ddt' has not yet been defined.

- lines 82-83: 'A further complication is that ILAR at 150° and YKA at 135° surround PKIKP with distinct patterns of other arrivals.' The implication of this statement is not clear to me. Can this be clarified?

- lines 155-156: 'some pairs that the model predicts to match do not'. Can this be quantified? How many, compared to pairs that are consistent with the model?

Renaud Deguen

Wednesday, November 29, 2023

Reviews annotated with comments

Referee #1:

I very much like the results on the inner core in which the authors show that the inner core moved forward and then backward, re-occupying the same position twice. I have seen the research being presented at conferences, and I am happy to see it is being submitted for publication in Nature.

It is the most carefully performed seismological study of a number of similar recently published studies on the same theme of the inner core oscillating back-and-forward or having a variable rotation rate instead of simply rotating in the same direction with a constant wave speed as was initially suggested.

I think the methodology is sound and the authors are very honest about what their data can and cannot resolve. I just have one question regarding the seismic data. When looking at their data in Figure 2, I did notice that in panel B, the first few swings of the PKIKP arrival (around -8 to -6 seconds) are in fact the same for all three events, and the main difference is in the train after the main PKIKP at -6 to 0 seconds. Why would the difference only be in the later part and not in the main PKIKP arrival?

This is a general observation for all the differences, and we have investigated it by computing simple 2-D scattering calculations. It turns out forward scattering through a changing medium produces greater changes the further one progresses into the coda, see upper right frame of co-author Ruoyan Wang's 2023 AGU Fall Meeting figure below. Pang and Koper (2022, Figure 7) have already made this point with more limited simulations. We've added two sentences describing this (lines 100-103).

Still, the abruptness of the transition from “no change” to distinctly different in the coda is surprising in some cases. It might also be the case for some repeaters that some parts of the IC have only radial structure, so that only energy that has traveled sufficiently far from the geometric ray path to reach places with lateral variation change when the IC rotates.

I guess my main hesitation with the current paper is that the authors should much more clearly state how their results are new and different from previous work especially their own publication (Wang & Vidale, Sci. Adv. 2022) and Pang & Koper (2022) and Yang & Song (2023). In fact, I think these studies are very different and the results in the current manuscript are far superior compared to the other three papers. But that should be more obvious from reading the paper.

This paper makes three advances. (1) We demonstrate the effectiveness of charting inner core motion by repeated waveforms. (2) A slow reversal since ~2003 is unambiguously resolved, which should end consideration of a six-year oscillation and models with no motion. (3) Early super-rotation is shown to be several times faster than later sub-rotation, so motion is not just a sinusoid, which would be expected from simple gravitational oscillation models. We refine the end of the abstract to make this clearer.

On a smaller note, I also would like to see some more discussion and comparison with normal mode estimates of inner core superrotation, which also had trouble finding a consistent rotation (work by Laske and others). Would it be possible to reconcile their proposed model with the normal mode estimates? Some other studies have suggested that the inner core may in fact only be rotating very slowly, at rates of 1 degree per million years or even less (work by Waszek and others). Would it also be possible to reconcile the proposed model with much slower rates over a long time?

Gabi Laske, to my knowledge, has three data points on the question of rotation. (1) An initial paper with Guy Masters found $0^\circ \pm 1^\circ$ per year motion. (2) She refined the analysis to obtain $0.13^\circ \pm 0.1^\circ$ in the paper we cite as reference 10. (3) She last told me about a month ago that there were no new results yet. Her best results thus suggest IC turned $\sim 2^\circ$ over several decades up to ~2000, not inconsistent with our and Song's models. I do not think there is more of relevance to add, especially given Nature's length limits.

Waszek's very slow motion is derived from the apparent tilt of near-vertical lateral contrasts in seismic wave velocity in the upper hundreds of km of the IC. Such a tilt would derive from much longer period drift of the IC, given its geologically slow growth rate. It should be compared to whether an oscillating IC continually oscillates with a fixed equilibrium point or alternatively trends over the millions of years over which the IC grows. We have no information about the very long-term drift of the IC from 40 or even 100 years of motion, so there is no inconsistency in the presence of a tilting interface with our models. Only if there were more space would mention of this feature be warranted.

Referee #2:

This manuscript presents PKIKP coda waveform changes (which is claimed by the authors as PKIKP waveform changes) recorded in two seismic arrays in Canada (YAK) and Alaska (ILAR). The authors claim to present the most definitive seismic evidence to date that the inner core has differentially rotated with respect to the rest of the Earth and the inner core has backtracked with the time of the reversal being around 2008. They go on and infer the speeds of inner core differential rotation before and after 2008, and conclude that the inner core super-

rotates from 2003-2008 and sub-rotates backward with a speed two or three times slower through the same path.

No information is given for any relocation procedures and any relocation results of the doublets. Those doublets are presented with the catalog information. The doublet source comes mostly from the databases of several research groups, with some newly identified by the authors. In the doublet sources, only those from Yao et al. (2009) had the relocation details and results. The source of the majority of the doublets in the list comes from Yang and Song's group. The doublets from Yang and Song's group were found to contain significant relocation errors (Zhang and Wen, 2023). The doublets need to be carefully relocated and need to be checked whether they are really close to each other. The temporal travel time changes need to consider the effects of the doublets' relative locations and origin times. Absent from the detailed analysis of doublet relative locations and their effects on the temporal change, the seismic results presented in the manuscript are questionable.

We respond to the issues of whether all timing and waveform changes arise from ICB changes comprehensively below. These are the main factors that are in play - (1) small differences in locations of events in a repeating pair, (2) the potential for pervasive clock errors, and (3) whether IC-reflected waveforms change all at once below.

My following comments are based on the assumption that all the results stand after an analysis of doublet relocation. I do not see major changes in PKIKP waveform as the authors claim. Most of the changes occurred in the PKIKP waveform coda after the main phases (-1 s, -3 s and -7 s in Fig. 2a, and -2 s and -6 s in Fig. 2b). So, I would just use PKIKP coda changes in the subsequent comments. The distinction is needed as the PKIKP coda originate from a very different location of the inner core than the sampling region of the PKIKP waveforms.

See our comments on the same observation above for Reviewer #1. Synthetics verify that later arrivals would change much more due to IC motion than the first arrival. An interesting observation, and we are exploring some of its implications in ongoing research, but it is in line with our conclusion.

Comment on the authors' definitive evidence of inner core backtracking -

The major evidence the authors claim that the inner core differentially rotates and backtracks to the original location is that the PKIKP coda change patterns between some triplets (three repeated earthquakes) and multiples (multiple repeated earthquakes). I use a triplet (events A,B,C, sequenced progressing in event occurring time) as an example for the discussion (Same logic applies to the multiples). The PKIKP coda exhibit changes from A to B, but no change from A to C. The authors interpret the change from A to B as the evidence that the inner core differentially rotates and, especially, no change from A to C as the inner core differentially rotates and backtracks to the initial position of A from B to C. The authors state that there is "the most definitive evidence to date that the IC is moving, and specifically that it is slowly and smoothly rotating on a reversible path."

PKIKP coda waveform changes in these arrays were well documented from the events occurring in the South Sandwich events (Figs. 2, S12-S16, S19-S22, Yao et al, 2019). Some sources of the PKIKP coda change were also well studied with a combined dataset from the two arrays used in

this manuscript (YAK and ILAR) and three more seismic arrays (IMAR, BMAR, BCAR), with the waveform changes identified to be caused by a seismic scatterer near the inner core surface (Figs. 3, S17-1S18, Yao et al. 2019) that subsequently disappeared (Fig. 4, Session 3.1, Yao et al. 2019). Similar behaviors of the changing seismic scatterers were also well documented in Yao et. al. (2015) that the temporal change of the inner core surface is episodic, rapidly migrating, and alternately enlarged and shrunk.

Scattered wavefields from events even only 100 km apart in our dataset do not show a common pattern of changes in scattering or even just the scattered arrivals themselves - it is unlikely that changes in coda phasing from repeaters many thousands of km apart would clearly reveal the location of individual scatterers. These are continuously interfering scattered waves, probably from pervasive heterogeneity, not scattering from individual bumps that are rare and widely spaced. Pang and Koper found, from simulations, that simple models of heterogeneity do not allow such tracking of peaks for widely placed stations.

The above-mentioned triplet waveform observations the authors presented can be simply explained by a mechanism that was well established in the observational evidence in the above literature: a disappearing scatterer. That is, an episodic change of the inner core surface appears around or before the time of even B and disappears after event B but before event C. The emergence of an episodic change of the inner core surface around or before event B would create waveform changes between AB, and the disappearance of the scatterer, or the recovery of the inner core surface, after event B and before event C would make waveforms dissimilar between BC and similar between events A and C. Or broadly, any localized waveform-changing scatterer that occurs around or before event B and subsequently disappears after event B but before C could explain the authors' observation. Inner core differential rotation is not needed. Inner core backtracking is just one of the possible ways to explain the data. It is not a piece of definitive evidence that the IC is moving or backtracking.

Yes, any one change and reversion have many possible interpretations, including that the middle event somehow was not a good repetition or showed a temporary change, although that possibility is minimized by the good agreement required for non-IC phases. Our conclusion is built on the pattern seen over time with widely distributed raypath segments within IC, and the lack of any pattern in space that would indicate changes in only limited areas.

The authors' explanation of those triplet data as the inner core backtracking is contradicted by the seismic evidence presented in the manuscript as I will comment later. However, the rule of science would require the authors to address the following issues before continuing to present their favor hypothesis of inner core differential rotation.

For the objectivity of science, the past research results of Yao et al. (2019) and the above existing candidate interpretation need to be acknowledged. Alternatively, the authors should demonstrate the results in Yao et al. (2019) are not correct before claiming "the definitive evidence" for the inner core differential rotation. In the studies of the inner core differential rotation (same in this manuscript), Yao et. al. (2019) was acknowledged providing an "alternative explanation". The temporal change of the inner core surface was shown to be a required interpretation for the observed temporal change of PKiKP phases in the earlier studies (Wen 2006, Yao et. al. 2021). Yao et. al. (2019) presented three lines of seismic evidence that are contradictory to the hypothesis of inner core differentiation. The subsequent published inner

core rotation papers never addressed those three lines of contradictory evidence. Neither does this manuscript. Only one piece of contradictory evidence is needed to refute a hypothesis. The authors need to address the three lines of the contradictory evidence in Yao et al. (2019) before continuing to present their interpretation.

Disagreement about whether the IC rotates at all based of differential time measurements centers on three disputes.

(1) Differences in repeater locations would affect interpretations of which phases have shifted when differential timing is observed to change. Dr. Song's group claims repeaters are co-sited to within 100-200m, Dr. Wen's group from careful relocations find differences up to about a km. In reply, Dr. Song has raised the 2nd issue, clock errors, which he argues preclude accurate relative locations. To me (JV), this issue remains unresolved.

(2) Pervasive small clock errors on GSN stations would affect interpretation of relative location and also absolute timing, which also affects whether IC-penetrating and/or ICB reflecting waves are shifting in time. Dr. Song's group worked with Albuquerque Seismology Lab scientist Adam Ringer to argue for errors greater than 0.03s on 10-20% of global stations in the last few decades (Yang et al., 2022). A recent SRL paper has corroborated some of these clock errors (Davis et al., 2023), and we found the different surprising and indisputable set of greater than 0.1 s clock errors for YKA described in the paper. There will be a poster on the topic of Dr. Song's claim of clock errors by Zhang & Wen at AGU next week (S11F-0320). To me (JV), this issue also remains unresolved.

(3) Interpretation of whether the ICB-reflected or IC-transmitted arrivals change waveform in the cases that they overlap results in a similar ambiguity that has been argued. As we note below, we see no clear changes in PKiKP, but even if present, PKiKP changes would not preclude rotation.

Our tracking of absolute time has been the most careful to date, and the resulting good alignment of the repeated waveforms does helps us judge waveform similarity. However, we still have difficulty identifying absolute timing changes because the YKA array has clock errors, as we documented, and the ILAR array has only at most tiny shifts on PKP_{df} arrivals that change waveform considerably, so we are not presenting estimates of ddt changes.

Our analysis neatly circumvents the controversies because we are only using repeaters that are known to highly correlate for non-IC phase waveforms for at least during the 15-s window (mostly longer than the PKiKP and its coda window for the analysis in the manuscript) that we use. We can confidently ascertain when the repeaters show similar enough waveforms to compare or not, and it does not matter if there are clock errors of a few hundredths of a second or differences in location of less than a km, which only affect timing and do not affect the waveforms. Changes in the combination of PKiKP and PKiKP that reverse are most naturally explained the same way as changes in only PKiKP, they are consistent with the IC rotating back to the same position without changes in the local ICB.

To summarize, there are several claims of ICB changes by Dr. Wen's group, which could be correct, but which are contested by Dr. Song's group. However, even if they are correct, they do not require that the IC is so fixed that our model of rotation is contradicted. Changes in any one repeater pair could be changes in the ICB, however it is the pattern of several dozen systematic progressive changes and reversions over many years that decisively favors our model for IC rotation.

In addition, the authors need to demonstrate the theoretical plausibility of their favor interpretation. For authors' proposal to explain the data, following work needs to be done in order to make the proposal to be even theoretically plausible:

- 1) Authors need to identify the actual seismic structure that causes the temporal change of the coda waves from A to B.
- 2) Authors need to demonstrate that the identified static structure could cause the temporal changes of the coda waves based on synthetic modeling. As the Fresnel Zone is large for the seismic waves used in the study (see an example of the Fresnel Zone in Fig. 1 in Tian and Wen (2023)), a slight change of a static seismic structure would not be able to produce much travel time and waveform change. Previous synthetic tests indicated that a static structure would need to be moved over a large distance to see the seismic effects, requiring an unreasonable super-rotation of the inner core (see an example of synthetic tests in Fig. 8, Yao et al. 2019). The shortest time of waveform change observed in this manuscript occurs in a time scale of about 9 months (Fig. S2). The authors need to demonstrate that a moving of the static structure identified can produce the waveform changes observed.

While the inner core differential rotation studies have always illustrated proposals by beautiful cartoons, the above two points need to be synthetically tested. Before the above two points can be demonstrated, the inner core differential rotation cannot even be regarded as physically plausible.

Comment on the contradiction of inner core backtracking to the actual data and the actual analysis in the manuscript -

The authors only analyze the data points of the doublets that exhibit no waveform change and have a time span of greater than 10 years, and discard those of the doublets that have a time span of less than 10 years and those that exhibit temporal changes. The authors state "as just a few years separation throughout the entire period apparently does not involve enough IC motion to always change the waveform". There are two problems with this approach: 1) the statement is not justified and 2) even in the selected dataset of the analysis, there are many contradictory data points and those contradictory data points are discarded.

- 1) The statement "as just a few years separation throughout the entire period apparently does not involve enough IC motion to always change the waveform" is not justified based on the actual observations presented in the manuscript and the inferred time spans of doublet waveform change based on their conclusion.

From the actual observations:

Many doublets exhibit waveform changes when their time span is less than 10 years. Just some examples from Fig. S1 only, these are the doublets that exhibit waveform changes but have a time span of less than 10 years: P41, P64, P99, 101, P43, 152, P88, PB3, P73, P10, 117, P14, P97, P28, P08, P12, P09, P2, P37. The shortest time span of the doublet that exhibits waveform change is event pair 116 in Fig. S2, which is 9 months and 8 days.

Yes, the data agreement with the model is not perfect. As the reviewer comments, perhaps not all pairs are perfect repeats, which is why we also checked that the most similar subset of repeaters

shows the same patterns. Also, some pairs were listed as only an intermediate level of match because of the noise level, even though the same noise level would allow clear identification of arrivals that do not match. Also, not every change is necessarily rotation. However, dozens of pairs of waveforms match in accord with our model, which has much in common with Song's models, which are mostly independently derived from ddt, and are unlikely to be explained by alternatives.

From the inferred time spans that must create waveform changes based on the author's differential rotation interpretation:

I just note one in Fig. S1 for example:

P109 (Red, no change) (06/06/03 – 08/27/21) vs. P140 (Blue, change) (05/26/03 – 05/30/22)

P140 occurs 10 days earlier and about 9 months later than P109. As anything occurs in the time period of P109 would be recovered based on the author's theory that the inner core backtracked to the original position, the waveform changes must occur in about 9 months in the time spans that the two doublets do not overlay.

The above observations and inference clearly show that the time scale of detectable temporal change in this dataset is as short as 9 months and that the author's discard of the data points of the doublets that have a time span of less than 10 years is clearly unjustified.

2) Even in the selected data points of the doublets that have a time span of more than 10 years, which the authors presume representing the backtracking of the inner core to the initial position and use to infer the time of the inner core reversal and the super-rotation speeds before and after the reverse, many data points are directly contradicted by the observations in the other array. From Figs. S1 and S2, I list a few here:

P33: Red (no waveform change) in ILAR, Blue (waveform change) in YAK. In other words, ILAR indicates that the inner core backtracks to its original position and the data point was used to infer the backtracking (based on the authors' theory and analysis). But YAK indicates that the inner core does not backtrack to its original position as it exhibits waveform changes, and this contradictory data point is discarded.

P31: Red in ILAR, Blue in YAK.

P78: Red in ILAR, Blue in YAK.

P68: Red in YAK, Blue in ILAR.

P51: Red in YAK, Blue in ILAR.

All the above Blue contradictory data points are discarded in the analysis.

Putting back into analysis the data points of the doublets that have a time span of less than 10 years, we now do not see any consistent pattern of possible reverse of the inner core [e.g., Fig. S1]. Instead, we have various "modes" of inner core backtracking with the reversals occurring at different times and depending on which doublet data points you use. The data points for the reversal are contradicted by the waveform changes of other doublet data in the same time

periods. We also see no evidence of reversal from the doublets occurring in the latitude range between P21 and P08 [Fig. S1] from YAK and that between 140 and P68 from ILAR [Fig. S2].

More contradictory data points emerge between the arrays in the doublets that have a time span of less than 10 years [Figs. S1, S2].

P41: Red in ILAR, Blue in YAK.

P43: Red in ILAR, Blue in YAK.

P39: Red in ILAR, Blue in YAK.

P116: Red in YAK, Blue in ILAR.

P31: Red in ILAR, Blue in YAK.

Clearly, the interpretation of the no temporal waveform change between the doublets or triplets as the result of inner core backtracking is not consistent with the seismic data, when we collectively and objectively analyze the seismic data. The seismic observations presented in the manuscript is a clear piece of seismic evidence that contradicts the hypothesis of inner core differential rotation.

We're not arguing that we explain everything in the literature claimed to arise from rotation, just that the data we present is consistent with and very well explained by rotation, and the inferred rotation similar to patterns previously resolved.

We're arguing for changes in a scattered wavefield arising from shifting the scattering material ten km or so. Shifting heterogeneity changes both focusing and arrival time, and hence interference between waves. The changes we see may be modeled as due to on the order of 1° IC rotation, although we are not explicit. 1° is about 20 km at the ICB. Pang and Koper (2022) have modeled waveform changes and found these rotations are plausible and interpreted some of these changes as corresponding to roughly to 0.5° rotation.

We have beamformed the scattered wavefield to ascertain that it comes from the expected direction and seems to arise within a broader area than the initial arrival. The changes are pervasive in the waveforms, not discrete blips, as mentioned above. This objection does not seem valid, in fact, the scatterers are likely not locatable isolated regions.

Yes, there is the pattern that events in the southern SSI are difficult to interpret for YKA, the changes are subtle. Also, to foreshadow our next paper, YKA shows some subtle changes that do not appear attributable to IC rotation. That does not preclude IC rotation. Sorry to be repetitive here.

In fact, the seismic results in the manuscript are also contradictory to the most recent high-profile conclusions of inner core differential rotation, including those of the authors of this manuscript.

Wang and Wei (Vidale, I think you mean) (2022) found that "the inner core subrotated at least 0.1° from 1969 to 1971, in contrast to superrotation of $\sim 0.29^\circ$ from 1971 to 1974" and proposed that the inner core oscillates in a 6-year cycle. Wang and Wei (2022)'s results were derived from the data points of the two groups of nuclear tests in a year span of 5 years, equivalent to one or

two data points in Figs. S1 or S2. No 6-year cycle signal can be found in the 33-year time span of the data in Figs. S1 and S2. If we randomly select two sets of the doublets in Fig. S1 or Fig. S2 (like one nuclear test group from 1969-1971 and the other nuclear group from 1971-1974”) and presume the inner core differential rotation as the interpretation, we can derive many “modes” of inner core differential rotation that are contradictory to the bulk of the observations.

The papers by us from nuclear tests show the IC probably sub-rotated 0.1° from 1969 to 1971 and certainly super-rotated 0.3° from 1971 to 1974. As this matched the timing and amplitude of IC motion in the six-year oscillation model of Chao, we favored that interpretation in our 2022 paper, but we are clearly not seeing a six-year oscillation in the last few decades, so we are no longer promoting that model.

Pang and Keith (Koper) (2022) concluded “We find that the inner core was nearly locked to the mantle before 2001 and after 2003 with relatively small motion about an equilibrium position. During 2001-2003, the inner core experienced a burst of differential rotation”. While their results were derived from a dataset that is also used in this manuscript, their conclusion is directly contradicted by the many BLUE doublet observations that have no time overlaps with 2001-2003 [Fig. S1] before 2017 (the year Pang and Keith (2022)’s doublet database ends).

Yes, our interpretations closely match those presented in Pang and Koper, and a bit surprisingly, the additional repeater pairs that we add (post 2017) do not match their earlier pattern so well. As the IC has only recently re-entered the position that apparently it had during the 2001-2003 interval in which Pang and Koper saw the greatest PKPdf waveform changes, it will be very interesting to watch the next few years of doublets, should the IC continue to backtrack through those positions, as seems very likely.

Yang and Song (2023) concluded “that all of the paths that previously showed significant temporal changes have exhibited little change over the past decade. This globally consistent pattern suggests that differential inner-core rotation has recently paused.” That conclusion is directly contradicted by the many BLUE doublet observations that started after 2010 (when the last decade starts).

Our model has the inner core slow and reverse around 2010, and only by continuing to analyze up to the most recent repeater pairs is change becoming very clear in recent years.

The reason that different groups or same groups at different publications obtained inconsistent results is that those inner differential rotation results were inferred from the seismic signals that are not related to inner core differential rotation and are intrinsically of another origin. Those seismic signals come from localized episodic temporal changes of the inner core surface that behave differently in differential geographic locations and in different time periods. They exhibit different temporal and spatial characteristics among differential seismic datasets. It is the misinterpretation of the seismic signals of the episodic localized changes of the inner core surface, coupled with the ignoring of the contradictory evidence, that resulted in many reported contradictory “modes” of inner core differential rotation.

We’ve simply concentrated on a single path, by far the best sampled path in the world, and found several dozen repeating pairs that have reverted to similar waveforms with the timing appropriate for a simple model of reversing inner core rotation. These changes do not indicate just a couple

of spots changing in the IC or on the ICB, the changes generally start within a second or two of the first arrival and remain different for the rest, generally up to 10 second, of the PKPdf arrival. The ~10-second coda can sample a much larger area within the heterogenous IC. The PKiKP coda (Wang and Vidale, 2022, EPSL) and normal mode (Laske and Masters, 2000) can sample larger regions within the IC or even the whole IC, respectively. Both observations can also demonstrate the IC rotation. Our model turns out to be somewhat like the model of Yang and Song (2023), albeit with the general pattern better resolved, with more details, and progressing further up to the present. Again, the localized signals cannot be used explain the general first-order pattern in our observation.

It is true and unexplainable that the papers of the inner core differential studies could ignore the contradictory evidence in the literature (papers after Yao et al. 2019), discard contradictory observations from the analyses (Yang and Song, 2022 vs. Tian and Wen, 2003), and make statements likely “unequivocally” without consideration of the alternatives and its contradiction to the other data (Yang and Song, 2022 vs. Tian and Wen, 2023). They could also discredit the other competing proposal with “erroneous claims” (Yang and Song, 2020 vs. Yao et al., 2021) or unreproducible results (Yang et al. 2021, vs. Zhang and Wen, 2023). Similar approaches are also adopted in this manuscript as I comment above. Another example is the statement in this manuscript that “Many PKiKP waves showed changes over the years, while no non-PKiKP phases resolvably changed in either arrival time or waveform, including IC reflected phases.”. While no results of doublet relocation were presented, the data experienced manual shifts, no identification of PKiKP phase was shown, and no mention of how the separation of PKiKP and PKiKP was made in the data, yet the authors make such a definitive statement that no change in arrival time in the IC reflected phases.

It would be an entirely different and interesting paper, as we have very many repeated arrivals of PKiKP in the data from ILAR to show for this path, and not even one shows a definite change in waveform nor timing. However, we’re glad to soften the offending sentence, as it does not play any role in the conclusions in this paper.

“no non-PKiKP phases resolvably changed in either arrival time or waveform, including IC reflected phases”

to

“we noticed no evidence that non-PKiKP phases changed ...” (Line 73-74)

It is also true that high-profile publications of inner core differential rotation and “exotic behaviors” of the differential rotation would capture one wave of media/public attention after another, as demonstrated in the past publications.

However, the ultimate outcome of ignoring the contradictory evidence, excluding the data that do not fit the hypothesis, and making a proposal without support of the physics of wave propagation will be the erosion of seismology as a credible branch of science and the destruction of seismologists as credible researchers.

Science advances by proposing the most likely explanation(s) for phenomena of greatest interest and import, and refining conclusions as evidence is tested and improved. Even now, it would not be surprising if our observations continue to be refined and models evolve.

In summary, while this manuscript presents interesting observations of waveform recovery across some doublets and a good dataset of temporal change of PKIKP coda waves, the interpretation of inner core differential rotation and backtracking is not supported by the bulk of the seismic observations presented in the manuscript. The subsequent results of the inferred inner core differential motions were obtained based on unjustifiable exclusions of the seismic data that do not support the hypothesis. The manuscript also failed to address the contradictory evidence to the hypothesis that was presented in the literature and in the manuscript, and the theoretical plausibility of the hypothesis. For these reasons, I do not recommend this manuscript for publication.

While it is not in my role as a referee to offer my view on the correct interpretation of the seismic data presented in this manuscript, I venture anywhere in case it is helpful. I think the authors have a unique dataset that sample one of the regions with intensive episodic changes of the inner core surface. Figs. S1-S2 clearly capture the spatial and temporal changes of a portion of the inner core surface that vary greatly in location and in time. A simple explanation exists to this seemingly contradictory and complex dataset. Each doublet captures a temporal change behavior at a particular spot of the inner core surface in the region during the occurring times of the doublet. A doublet with temporal change captures a temporal change of the inner core surface, a triplet or multiple with recovering waveforms reveals a temporal change of the inner core surface that has subsequently recovered, a doublet with contrasting temporal change patterns between the arrays (no change in one array and change in the other) represents a temporal change of the inner core surface that is more sensitive to one array and less sensitive to the other (due to location or the style of the change), and a doublet without change represents either a temporal change of the inner core surface that has subsequently recovered or no change of inner core surface. The dataset is a clear piece of observational evidence that contradicts the hypothesis of inner core differential rotation and is a unique set of seismic constraints that could illuminate a detailed picture of episodic spatial and temporal changes of a particular region of the inner core surface. One way or the other, I am confident that this correct version of the interpretation will be published in a scientific journal.

Interesting speculation. We did check whether the changes we present could be explained by a spatio-temporally propagating IC or ICB disturbance or just a localized change, without success. As I alluded to above (and we mentioned at least twice in the original manuscript), we think there might be changes visible in addition to rotation that reinforce interpretations of shallow IC changes, but that the rotation interpretation we present remains robust.

Wen, L. (2006). Localized Temporal Change of the Earth's Inner Core Boundary, *Science*, 314, no. 5801, 967-970, doi: 10.1126/science.1131692.

Yang, Y., and X. Song (2020). Origin of temporal changes of inner-core seismic waves, *Earth Planet. Sci. Lett.*, 541, no. 116267, doi: 10.1016/j.epsl.2020.116267.

Yao, J., D. Tian, L. Sun, and L. Wen (2021). Comment on “Origin of temporal changes of inner-core seismic waves” by Yang and Song (2020), *Earth Planet. Sci. Lett.*, 553, no. 116640, doi: 10.1016/j.epsl.2020.116640.

Yang, Y., and X. Song (2022). Inner Core Rotation Captured by Earthquake Doublets and Twin Stations, *Geophys. Res. Lett.*, 49, no. 12, e2022GL098393, doi: 10.1029/2022GL098393.

Tian, D., and L. Wen (2023). Comment on “Inner Core Rotation Captured by Earthquake Doublets and Twin Stations” by Yang and Song, *Geophys. Res. Lett.*, 50, no. 15, e2023GL103173, doi: 10.1029/2023GL103173.

Yang, Y., X. Song, and A. T. Ringler (2021). An Evaluation of the Timing Accuracy of Global and Regional Seismic Stations and Networks, *Seismol. Res. Lett.*, 93, no. 1, 161-172, doi: 10.1785/0220210232.

Zhang, X. and L. Wen, (2023). Problematic Reported "Prevailing Clock Errors in Seismic Stations": Comment on "An Evaluation of the Timing Accuracy of Global and Regional Seismic Stations and Networks" by Yang et al. (2021), <https://agu.confex.com/agu/fm23/meetingapp.cgi/Paper/1349595>).

Yao, J., L. Sun, and L. Wen (2015). Two decades of temporal change of Earth's inner core boundary, *J. Geophys. Res.*, 120, no. 9, 6263-6283, doi: 10.1002/2015JB012339.

Yao, J., D. Tian, L. Sun, and L. Wen (2019). Temporal Change of Seismic Earth's Inner Core Phases: Inner Core Differential Rotation or Temporal Change of Inner Core Surface?, *J. Geophys. Res.*, 124, no. 7, 6720-6736, doi: 10.1029/2019JB017532.

Wang, W., and J. E. Vidale (2022). Seismological observation of Earth’s oscillating inner core. *Sci. Adv.*, 8(23), eabm9916.

Pang, G., and K. D. Koper (2022). Excitation of Earth's inner core rotational oscillation during 2001–2003 captured by earthquake doublets. *Earth Planet. Sci. Lett.*, 584, 117504.

Yang, Y., and X. Song (2023). Multidecadal variation of the Earth’s inner-core rotation. *Nat. Geosci.*, 16, 182–187.

Referee #3:

The paper proposes a new way of using seismic multiplets to constrain the rotation of Earth's inner core (IC).

The analysis suggests that the rotation of the IC with respect to the mantle has changed direction around 2008. The authors base their conclusions on a detailed study of the waveform of seismic waves traveling through the IC (PKIKP), from a series of repeating South Sandwich Islands earthquakes (multiplets) from 1993 to 2023. The crucial observation is that in a number of multiplets, the waveform is observed to change with time before reverting to match the waveform of earlier events. The events with similar waveforms are interpreted as indicating that the IC has the same orientation with respect to the mantle, while different waveforms indicate different IC orientations. The analysis of all observed pairs of similar waveform events show a consistent pattern suggesting a change in IC rotation direction around 2008, as well as different rotation rates before and after 2008.

The results have important implications for the understanding of inner core dynamics and interaction with the core and mantle. The motion of the IC with respect to the mantle has been the subject of much debate. Robust observational constraints on IC rotation can provide key

information for understanding angular momentum exchange between mantle, outer core, and inner core, and provide constraints on the mantle/inner core gravitational coupling and IC effective viscosity.

This is to my knowledge the first time that the waveform of IC sensitive multiplets are analyzed and used in this way. I find these observations quite exciting, and their analysis elegant.

As far as I can judge (I'm not a seismologist), the seismological observations seem robust, but I will leave this question to more knowledgeable reviewers. The model is simple (this is one of the strengths of the paper), and the assumptions behind it are clear for the most part.

Very good understanding of the seismology for a non-seismologist.

The paper is clearly written and I have very much enjoyed reading it.

Apart from the suggestion detailed in my first point below, I have only very minor comments.

(1) While reading the paper, I have been wondering about how likely it is to sample the same IC orientation at two times picked at random, and about whether looking at the observations presented in the paper from a probabilistic point of view might give any useful information.

As an illustration, I did the following simple calculation, which perhaps will be of some use to the authors.

Let's denote by φ the rotation angle of the IC with respect to some reference, by $\Delta\varphi$ the difference of orientation of the IC between two events, and by $\Delta\varphi_c$ a 'correlation angle' defined as being the change of φ below which no noticeable change in waveform can be found (i.e. the two waveforms are similar if $\Delta\varphi < \Delta\varphi_c$). If multiplied by the radius of IC, $\Delta\varphi_c$ would give a length scale characteristic of IC internal structure (a 'correlation length').

It is found by the authors that roughly one fourth of the studied doublets happen to have similar waveforms (57 similar out of 200 doublets).

This could be interpreted as meaning that the probability, when taking a pair of observation times at random, of sampling the same orientation φ within an interval of width $\Delta\varphi_c$ is about 25% ($\sim 57/200$). Or, equivalently, that there is a probability of about 25% to find $\Delta\varphi < \Delta\varphi_c$ for a random pair of events. Maybe some useful information can be extracted from this observation.

To be explicit, I have assumed that the angle φ evolves with time t as $\varphi = \varphi_0 (t/T)^2$ for t between $-T$ and T , where $2T$ is the length of the observation period, and φ_0 the amplitude of change of φ over this period. $t=0$ corresponds to the reversal time. This is a 2nd order Taylor expansion around the reversal time, assuming a symmetrical behaviour around it (it can easily be modified to account for the observed asymmetry around the reversal time). If we assume the rotation rate of the inner core to be ~ 0.1 deg/year, and take $T \sim 15$ years to be consistent with the data of the paper, one gets $\varphi_0 \sim 1.5$ degree.

With these assumptions and a simple python script, I have drawn a random set of N pairs of times within the observation period, calculated the difference of orientation $\Delta\varphi$ for each pair of times, and then calculated the cumulated density function(CDF) of $\Delta\varphi$. The resulting CDF shows that the 25% of time pairs having the smallest orientation difference are such that $\Delta\varphi/\varphi_0$ is

smaller than about 0.1. In other words, 25% of events pair would have similar waveforms if the correlation angle $\Delta\varphi_c$ is about $0.1*\varphi_0$. With $\varphi_0 \sim 1.5$ degree, this gives $\Delta\varphi_c \sim 0.15$ degree. Converting this into a length scale by multiplying by the radius of the inner core gives a correlation length of ~ 3 km.

This estimate clearly depends on the assumed form of the $\varphi(t)$ function, on the parameters values I used, and, crucially, on whether the number of analysed event pairs is large enough to give a reliable estimate of the probability of finding similar waveforms. But at least the value obtained for the correlation length is not crazy, which to me gives additional support to the author's interpretation of their observations. And perhaps it is worth investigating further to check whether the value I have obtained for the correlation length is robust. If so that would give an additional measure of IC heterogeneity.

Anyway, this is really just a suggestion, which the authors are free to follow or not.

Excellent thought, which develops a possible framework (which we had not yet crystallized in such detail) for synthetic seismograms that we are calculating to understand the pattern of loss of correlation and time shifts and how they depend on time lag into the coda. But this extension would be most appropriate with the next paper rather than complicating this one.

(2) Another point, which is somewhat related to point (1):

I would think that the interpretation of the data discussed in the paper depends to some extent on assumptions on the spatial structure of IC heterogeneities. For example, if the IC is overall homogeneous with only very localised heterogeneities, then finding twice the same waveforms should be the norm, and would not necessarily indicate that the IC is in the same position. I would therefore think that an implicit assumption of the interpretation is that heterogeneities are evenly distributed, or that the IC properties vary gradually within the IC. I guess this interpretation is supported by the observations and others, but maybe this should be discussed?

This is a point deserving discussion. Yes, our cartoon is an inner core with somewhat distributed scattering, at least around this ray path of SSI \rightarrow COL, which was originally selected by Song 1996 due to visible changes. This model is supported by Wang and Vidale (2022, Nature Geoscience) in which back-scattering is broadly distributed, with a couple of notable concentrations. Indeed, the ILAR PKPdf observations are uniformly changing whenever the IC is inferred to have moved. The YKA observations are more complicated - ray paths from the southern SSI show less change than from the rest of the SSI. YKA arrivals are also a mix of PKPdf and IC reflections, with somewhat unknown relative amplitudes. Plus, there are weak hints that YKA sometimes changes even when the IC is in the same position. So YKA changes are harder to interpret. Yet another wild card is that many other paths have been found to not change, although it is usually measured with just ddt rather than waveforms. And another complication is that the scattering Fresnel regions contributing to the later coda becomes large after 5 or more seconds, so overlapping for most of the event pairs in this paper.

I hate to say it, but additional discussion would take a fair bit of space but not greatly clarify these particular issues, it would just underline the complexity of interpretation.

Additional remarks :

- line 20-21: 'The pattern of matches shows that the IC super-rotated from 2003 to 2008, and then from 2008 to 2023 sub-rotated'

If I understand correctly, the observations presented in the paper do not allow to differentiate between super-rotation and sub-rotation (in fig 3, taking fig 3b upside down, i.e. having phi increasing and then decreasing, has no effect on figure 3c) but only show that the polarity of motion has changed. The interpretation in terms of super-rotation and the sub-rotation comes from the literature. The sentence of the abstract should make this clear, either by only mentioning the change of polarity, or by mentioning the use of previous results from the literature.

This is correct. While an accurate interpretation would come from reading the paper and understanding the seismology, we tried to make it more clear by changing

The pattern of matches shows

to

The pattern of matches, in concert with previous studies, demonstrates (Line 21)

- line 47: the part of the sentence in between parenthesis ('and mostly rotation only during') is not clear to me.

Fair point, we changed

0.5° IC rotation during (and mostly rotation only during) that period²⁶.

to

0.5° IC rotation during that period²⁶ and much less rotation at other times. (Line 40)

- line 58: I believe 'ddt' has not yet been defined.

Oops, correct, we changed

claims of ddt change

to

changes in the time difference between core phases (ddt) (Line 60-61)

- lines 82-83: 'A further complication is that ILAR at 150° and YKA at 135° surround PKIKP with distinct patterns of other arrivals.' The implication of this statement is not clear to me. Can this be clarified?

We changed this to

present PKP waves with distinct patterns of timing and amplitude of PKIKP and PKiKP, and interference with other core phases. (Line 85-86)

- lines 155-156: 'some pairs that the model predicts to match do not'. Can this be quantified? How many, compared to pairs that are consistent with the model?

It is fair to request correlation coefficients, and we are working on this, however, this would not be simple to construct nor present. The distinct 137° vs 150° arrival patterns, noise levels, and correlation vs time delay into coda patterns, would require many free parameters. We prefer to offer simply the visual results and access to the waveforms for others to re-examine the patterns. Our model makes clear predictions for (the ample) independent PKP data for other regions.

Renaud Deguen

John Vidale and the rest of the authors.

John E. Vidale
jvidale@usc.edu, 310-210-2131
National Academy of Science member

Reviewer Reports on the First Revision:

Referee #1:

The authors have clearly answered all my questions, which has only further strengthened my advice to publish without any delay. The study is sounds, well performed and the seismological analysis has been very careful.

Arwen Deuss

Referee #2:

The authors addressed (some of) my concerns with statements. Most of the statements are not relevant to the issues I raised (for example, invoking the "clock error" debate , which has nothing to do with the three lines of contradictory evidence in Yao et al. (2019), does not address those three lines of contradictory evidence to the hypothesis of inner core differential rotation) . All statements are made without any supporting results of analysis.

A side note: Contrary to the authors' claim, no corroboration was ever presented in Davis et al. 2023 for the "clock errors" of the OBN and AAK stations in all the doublets claimed by Dr. Song's group. Davis et al. 2023 can be accessed here: <https://doi.org/10.1785/0220230174>. Dr. Song's group claim can be accessed here: <https://doi.org/10.1785/0220210232>. Dr. Wen's group's AGU abstract can be accessed here: http://geophysics.geo.sunysb.edu/wen/Reprints/ZhangWenAGU23_S11F-0320.pdf

The first-order feature is not the selected few that the authors use to promote the model (roughly 20 data points out of 143 repeaters x 2 arrays). It is in Figures S1 and S2, and that is contradictory to the authors' model. Even in the selected repeaters the authors use, the model cannot explain all the data (Fig. 4, keep in mind the time scale of observable change is 9 months) and is contradicted by the data between the arrays from the same selected repeaters, as I mentioned in my early review.

Regrettably, none of the issues I raised in the first round of the review was addressed. All the issues I raised remain true to this revised manuscript. Rather than relisting those issues, I copy and paste the summary of my last review here, which remains to be my assessment of this revised manuscript.

"In summary, while this manuscript presents interesting observations of waveform recovery across some doublets and a good dataset of temporal change of PKIKP coda waves, the interpretation of inner core differential rotation and backtracking is not supported by the bulk of the seismic observations presented in the manuscript. The subsequent results of the inferred inner core differential motions were obtained based on unjustifiable exclusions of the seismic data that do not support the hypothesis. The manuscript also failed to address the contradictory evidence to the hypothesis that was presented in the literature and in the manuscript, and the theoretical plausibility of the hypothesis. For these reasons, I

do not recommend this manuscript for publication."

Referee #3:

I am satisfied with the answers of the authors to my comments. This is a very nice and interesting paper, which in my view deserve to be published in Nature.

Author Rebuttals to First Revision:

We appreciate the thorough and deliberate reviews, which is sensible for the controversial topic of inner core temporal change, which has such subtle data trends to parse.

With regard to the steadfast objections of reviewer #2, he argues two main points,

1. We didn't relocate the earthquakes to determine the distance between events in each repeating earthquake pair.
2. While we do present 20 or so matching waveform pairs, there are many pairs whose waveforms do not match the expected pattern.

In response to the first of reviewer #2's misgivings, we'd elaborate that indeed estimating the slight differences in location between the earthquakes in each repeating pair, which we did not do, has been documented to be critical in measuring the temporal change in the time separation of core phases (ddt)^{9,10}. We avoid this complication by instead assessing waveform change, which is not sensitive to the very small time shifts that are the primary signal in ddt studies.

To corroborate this statement, Figure S6a in Pang and Koper (EPSL, 2022) using the locations from the relocated repeating earthquake catalogs in Yao et al. (2019) and Yang and Song (2020) shows that there is no clear relationship between the mislocations and the waveform similarity of the repeating pairs. The waveform changes in our study are not caused solely by small mislocations of the repeating pairs.

With regard to the second point, the paper already makes the points clearly enough, in our view, but we'll expand on it a bit here. We are basically seeking earthquake pairs whose waveforms match better than expected on the basis of the general level of change seen in the PKIKP phases over time, which varies along the Sandwich arc.

There are several ways to make the waveforms different - movement or other change in the inner core, a difference in focal mechanism, a larger than expected differences in source location, noise levels that preclude confidence that the waveforms match. In fact, we are working on a manuscript that some more subtle waveform changes may indeed be due to changes other than rotation of the inner core.

However, in order for waveforms that are expected to change, to instead match, a lack of inner core change is necessary. This is especially true for the cases in which the PKIKP is seen to change and then change back. So our analysis focuses on the these matches, only noting and not discussing the repeating pairs whose waveforms that changed. The pattern of matches makes a strikingly clean pattern that is in line with the recent models of Xiaodong Song, which are mostly derived from controversial ddt measurements.

It is true one could make a model involving only non-rotational changes in the inner core or ICB, however it would have to greatly disturb the paths along the entire range of latitudes along the Sandwich Island in a synchronous way, with the years matching lining up as shown in Figure 4. We also consider that, among many other papers, our papers on inner core scattering from nuclear test pairs show the inner core was clearly rotating in the early 1970s, so rotation is the most plausible hypothesis for these changes in the last few decades.

Reviewers #1 (Arwen Deuss) and #3 (Renaud Deguen) raised no issues.